



**Insoluble lipid film mediates the transfer of soluble saccharides from the sea to**
**the atmosphere: the role of hydrogen bonding**
Minglan Xu, Narcisse Tsona Tchinda, Jianlong Li, Lin Du*
Environment Research Institute, Shandong University, Binhai Road 72, Qingdao,
266237, China
Correspondence: Lin Du (lindu@sdu.edu.cn)





**Abstract**
Saccharides are a large group of organic matter in sea spray aerosol (SSA). Although
they can affect climate-related properties of SSA, the mechanism through which
saccharides are transferred from bulk seawater to the ocean surface and ultimately into
SSA is still debated. Here, the transfer of small soluble saccharides was validated and
quantified using a controlled plunging jet sea spray aerosol generator to better
understand the wide range of particle properties produced by natural seawater mixed
with model organic species, glucose and trehalose. Data show that both soluble
saccharides can promote the production of SSA particles. Conversely, the role of the
insoluble fatty acid film on the surface greatly reduced the production of SSA. The
resulting inorganic-organic mixed particles identified by the transmission electron
microscope (TEM) showed typical core-shell morphology. Langmuir model was used
to parameterize the adsorption and distribution of saccharide into SSA across the bubble
surface, while infrared reflection-absorption spectroscopy (IRRAS) combined with
Langmuir isotherms were undertaken to examine the effects of aqueous subphase
soluble saccharides on the phase behavior, structure and ordering of insoluble lipid
monolayers absorbed at the air/water interface. Changes in alkyl chains and headgroups
structure of mixed fatty acid monolayers under different saccharide concentrations in
aqueous phase were reported. In seawater solution, the effects of dissolved saccharides
on the ordering and organization of fatty acid chains were muted. Hydrogen bond
analysis implied that soluble saccharide molecules displaced a large amount of water
near the fatty acid polar headgroups. Saccharide-lipid interactions increased with



increasing complexity of the saccharide in the order glucose < trehalose. Our results
indicate that the interaction between soluble saccharides and insoluble fatty acid
molecules through hydrogen bonds is an important component of the sea-air transfer
mechanism of saccharides.
**1 Introduction**
Sea spray aerosol (SSA) represents the major source of aerosol particle populations
and significantly impacts the earth's radiation budget, cloud formation and
microphysics by serving as cloud condensation nuclei (CCN) and ice nuclei (IN), and
microbial cycling (Bertram et al., 2018; Partanen et al., 2014). The formation of SSA
particles is strongly influenced by the uppermost sea surface microlayer (SML), which
is a thin layer of 1–1000 μm thickness formed due to different physicochemical
properties of air and seawater (Wurl et al., 2017). Beyond sea salt, the ocean surface
contains a fair amount of organic matter (OM) mass fraction, covering carbohydrates,
lipids, proteins, humic-like, intact phytoplankton cells and fragments, fungi, viruses,
and bacteria (Van Pinxteren et al., 2020; Cunliffe et al., 2013). The SML is involved in
the generation of SSA, including their organic fractions by transferring OM to rising
bubbles before they burst into film drops and jet drops (Wang et al., 2017). When a
bubble reaches the water surface, destroying the surface membrane of the water, the
bubble bursts into many so-called film droplets. After the bubble film breaks, a jet of
water rising vertically from the ruptured bubble cavity forms so-called jet droplets.
During this process, the film drops (<1 μm) leading to the formation of submicron



particles are mainly OM-enriched compared to the larger jet drops (1–25 μm). Specific
organics, such as surface-active OM, are highly enriched in SML relative to bulk
seawater and contribute to surface film formation. They are mainly composed of
phospholipids, fatty acids, fatty alcohols, sterols and more complex colloids and
aggregates (Crocker et al., 2022; Van Pinxteren et al., 2022). Among these organics, the
saturated fatty acids are the dominant contributors (Cochran et al., 2016). The chemical
composition of SSA also depends on the physical properties of the breaking waves,
which determines the distribution of OM at the air-sea interface and how it is
transported from the SML into aerosol particles (Collins et al., 2014).
Surface-active biomolecules are preferentially transferred from marine surface water
into the atmosphere through the bubble bursting processes, forming a considerable
fraction of primary marine organic aerosols (Schmitt-Kopplin et al., 2012). Previous
measurements have shown that up to 60% of ocean particle mass can be organic (with
even a higher proportion for submicron particles), which exhibits a strong size
dependence(O'dowd et al., 2004; Russell et al., 2010). Spectroscopic evidence from
field-collected SSA particles indicates that the oxygen-rich organic fractions of
individual particles contains molecular signatures of saccharides and carboxylic acids
(Hawkins and Russell, 2010). For example, it has previously been observed that the
carbohydrate-like spectroscopic signatures account for 40–61% of the submicron SSA
organic mass (Quinn et al., 2014; Russell et al., 2010). A large portion of this mass is
attributed to saccharides that are transferred from seawater to SSA, and shows a certain
enrichment in SSA. Specifically, the high enrichment factor of carbohydrates was



calculated for supermicron (20–4000) and submicron (40–167000) particles relative to
the bulk seawater in the Western Antarctic Peninsula (Zeppenfeld et al., 2021).
According to previous laboratory studies, marine bacteria, divalent cations and protein
can affect the saccharide enrichment in SSA (Hasenecz et al., 2020; Schill et al., 2018).
However, a mechanistic and predictable understanding of these complex and interacting
processes in favor of saccharides found in marine aerosol particles remains largely
unexplored, despite their oceanic and atmospheric significance. More fundamentally,
the nature of marine organic aerosol particles composed of carbohydrates content and
their ability to act as CCN or IN are not entirely disentangled (Orellana et al., 2011;
Cochran et al., 2017; Wolf et al., 2019).
A variety of saccharides have been found ubiquitous in the ocean, including dissolved
free monosaccharides, oligo/polysaccharides, sugar alcohols, and monosaccharide
dehydrates, the composition of which depends on marine biological activity (Van
Pinxteren et al., 2012). Frossard et al. (2014) used the hydroxyl characteristic functional
group of atmospheric marine aerosols from Fourier transform infrared spectroscopy to
infer the contributions of different saccharides in SSA. It was found that the primary
marine aerosols produced in biologically productive seawater had stronger hydroxyl
group absorption peak characteristic of monosaccharides and disaccharides, while the
hydroxyl groups of seawater organic matter were closer to those of polysaccharides.
This suggests that larger saccharides may be preferentially retained in seawater during
aerosol production. Analysis of aerosol samples collected on the Western Antarctic
Peninsula also showed that not only polysaccharides but also a high portion of free

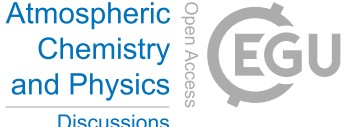

monosaccharides mainly composed of glucose, fructose, rhamnose and glucosamine
were present (Zeppenfeld et al., 2021). Raman spectroscopy was used to measure
individual SSA particles generated via wave breaking in a wave flume under algal
bloom conditions to get a deeper insight into their organic categories. It was reported
that 4%–17% and 3%–46% of sub- and supermicron particles show strong spectral
characteristics of free saccharides and short-chain fatty acids, respectively (Cochran et
al., 2017). However, current climate models largely underestimate the ratio of
saccharides in marine aerosols (Cravigan et al., 2020). It is urgent to clarify the
physicochemical mechanisms that drive free saccharides transfer to SSA.
A possible explanation for the SSA composition in saccharides involves the affinity
between soluble saccharides and insoluble surfactant monolayers already adsorbed on
the water surface, resulting in co-adsorption of the soluble saccharides (Link et al.,
2019b). This co-adsorption arises from non-covalent interactions and promotes the
binding of soluble organic matter to the surface with the headgroups of insoluble
Langmuir film. Previous studies have indicated that the presence of lipids or proteins
strongly enhances the surface adsorption capacity of saccharides, even for highly
soluble saccharides that do not adsorb individually at the air/water interface (Pavinatto
et al., 2007; Burrows et al., 2016). For example, recent studies have shown that simple,
soluble biomolecules such as phenylalanine and trehalose exhibit an affinity for lipid
films, altering membrane permeability and phase behavior (Perkins and Vaida, 2017;
Link et al., 2019a). A divalent cation-mediated co-adsorption mechanism was also
proposed to explain the enrichment of monosaccharide in laboratory-generated SSA



(Schill et al., 2018). Alternatively, saccharides can be bound covalently to larger, more
surface-active biomolecules, such as glycoproteins or lipopolysaccharides, which
attach to SML and are eventually transferred into SSA through bubble bursting at the
ocean surface (Estillore et al., 2017). Although different hypotheses have been proposed,
there is still debate about the more nuanced mechanisms that guide the sugar-lipid
interactions in the marine environment.
The present work aims to use a multipronged approach that combines bulk SSA
production experiments, Langmuir surface pressure-area isotherms and infrared
reflection-absorption spectroscopy (IRRAS) to examine the role of saccharides in SSA
production and the mechanism of saccharides transfer and enrichment from aqueous
solution into SSA. The study focuses on two small soluble saccharides that are
prevalent in seawater, glucose and trehalose, which are uncharged monosaccharide and
disaccharide, respectively. A plunging jet sea spray aerosol generator was used to
generate nascent SSA particles by artificially generating bubbles in seawater as a mean
of simulating sea spray production by breaking waves. This simulation helps evaluate
the impact of soluble saccharides as well as fatty acids on SSA production and particle
morphology. Langmuir isotherms provided abundant information for stability and
fluidity of monolayers, which were used to adequately describe the magnitude of
interaction effects between subphase soluble saccharides and surface insoluble
surfactants. Finally, IRRAS spectra provided molecular scale descriptions of monolayer
conformational information and allowed us to deduce the distribution of saccharide
species at the interface. By combining all the data, we propose a model of sea-air



transfer of marine saccharides through hydrogen bond interactions involved in surface
insoluble lipid molecules.
**2 Experimental section**
**2.1 Materials and solutions**
D-(+)-Glucose (Glu, powder, ≥99.5%) and D-(+)-Trehalose anhydrous (Tre, powder,
99%) were purchased from Aladdin. Stearic acid (SA, >98%, TCI), palmitic acid
(PA, >98%, Adamas-beta) and myristic acid (MA, ≥99.5%, Aladdin) were prepared in
chloroform (AR, ≥99.0%, Sinopharm Chemical Reagent Co., Ltd) at a final
concentration of 1 mM each. Figure S1 shows the chemical structures of the three fatty
acids used in this study. The respective fatty acid solutions were mixed at a molar ratio
of 2 MA:4 PA:3 SA to obtain a mixed lipid stock solution. All chemicals were used
without further purification. The natural seawater (SW) was collected from Shazikou,
Qingdao, China. Here, surface seawater was obtained from a pier on the coast by
dipping high-density polyethylene containers trough the seawater surface. The sampled
seawater was microfiltered through 0.2 μm polyethersulfone filter (Supor®-200, Pall
Life Sciences, USA) to remove large particles such as sediments, algae and bacteria.
The filtered seawater was used for SSA generation and as a filling subphase for
interfacial experiments. Different concentrations of saccharide-containing seawater
solution required in the experiments were obtained by dissolving different masses of
glucose or trehalose in the filtered natural seawater using mechanical stirring.



## 2.2 SSA production and collection

SSAs were produced using a plunging jet-sea spray aerosol generator (Figure 1). A physical drawing of the aerosol generation system can be found in Figure S2. The generator and its detailed operation principle has been described elsewhere (Liu et al., 2022). Briefly, the generator consists of a stainless steel (shipboard class, 316L) rectangular sealed container and a viewable glass window. The upper removable lid has ports for water inlet, purging air, and sampling. The purge air is supplied by a zero-air generator (Model 111, Thermo Scientific, USA) and the flow rate is controlled at 10 L min$^{-1}$. A peristaltic pump (WL600-1A, ShenChen) periodically circulates water from the bottom of the generator to the top nozzle through a Teflon tube with a pump speed of 1 L min$^{-1}$, creating a plunging water column that hits the seawater surface and entrains air into the bulk seawater. The bubble plumes extend approximately 15 cm down into the water, a moderate depth considering that the majority of the air being entrained in is located within about 50 cm from the sea surface (Hultin et al., 2010). When the bubbles rise to the air/water interface and burst, they generate SSA emissions. When studying insoluble surfactant effects, a concentrated solution of 1 mM mixed fatty acids in chloroform was added to the surface of the seawater solution. After the necessary fatty acids were added, only the sheath air flowed, allowing the chloroform to evaporate for 15 min and leaving only the surfactant on the surface. After pre-preparation for 15 min, the sheath air and peristaltic pump were turned on to produce SSAs. Prior to collection, SSAs were dried to a relative humidity of ~40% using a diffusion dryer. Thereafter, a scanning mobility particle sizer (SMPS, model 3936, TSI)





consisting of a differential mobility analyzer (DMA, model 3081, TSI Inc., USA) and
a condensation particle counter (CPC, model 3776, TSI Inc., USA) was used to measure
the particle size distributions and number concentrations, respectively. The particle size
distribution ranging from 13.6 to 710.5 nm was obtained at a sheath flow rate of 3.0 L
min$^{-1}$ and aerosol flow rate of 0.3 L min$^{-1}$. Dried SSAs were deposited onto 200 mesh
copper grids with carbon foil (T11023, Tilan, China) by a single particle sampler (DKL-
2, Genstar electronic technology Co., Ltd) to further characterize the particle
morphology.

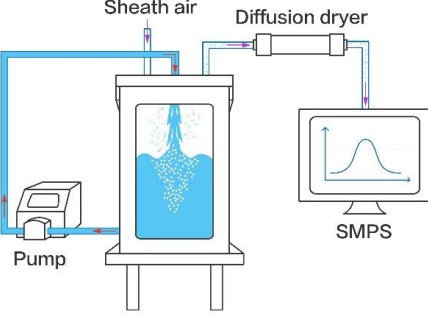


Figure 1. Schematic picture of the plunging jet-sea spray aerosol generator. The red
arrows represent the flow direction of seawater, and the purple arrows represent the
flow of gases and aerosols.
**2.3 Langmuir monolayer preparation and Langmuir isotherms**
The Langmuir trough setup has been described previously (Xu et al., 2021). Briefly, it
consists of a rectangular Teflon trough (Riegler & Kirstein, Germany) and two
moveable Teflon barriers whose movements are precisely controlled to achieve

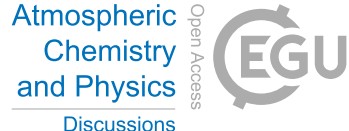
symmetric compression of the monolayer at the air/water interface. A Wilhelmy plate
attached to the pressure sensor was used to measure the surface pressure. Each 100 mL
subphase consisted of natural seawater, with varying amounts of glucose or trehalose.
Aliquots of mixed fatty acids stock solution were spread onto the subphase surface
dropwise with a glass microsyringe and 15–20 min were allowed for solvent
evaporation. The surface pressure ($\pi$), given by eq 1 and defined as the difference in
surface tension between the pure air/water interface ($\gamma_0$) and the monolayer covered
interface ($\gamma$) was monitored.
$$\pi = \gamma_0 - \gamma \qquad (1)$$
The barriers were compressed at 3 mm min$^{-1}$ and isotherm data were collected for
surface pressure $\pi$ (mN m$^{-1}$) versus area per molecule (Å$^2$). All experiments were
performed at $(22 \pm 3)$ ℃ and relative humidity below 65%.
**2.4 Infrared reflection-absorption spectroscopy measurement**
The polarization-modulation infrared reflection-absorption spectroscopy (PM-IRRAS)
is a mainstream spectroscopic method for in-situ characterization of Langmuir
monolayers at the molecular level. For IRRAS spectra, floating monolayers were spread
at the aqueous subphase and compressed to the desired surface pressure, and stopped
before obtaining the spectra. PM-IRRAS spectra were obtained using a Fourier
transform infrared (FT-IR) spectrometer (Bruker Vertex 70, Germany) equipped with
an external reflection accessory (XA-511). The interference infrared beam was set out
from FT-IR and polarized by a ZnSe polarizer to alternately generate s- and p-





polarization lights. They were then continuously modulated by a photoelastic modulator
(PEM-100) at a high frequency of 42 kHz to measure the spectra of both polarizations
simultaneously. The infrared beam was focused onto the Langmuir film through a gold
mirror, and then a portion of reflected light was directed onto the liquid nitrogen-cooled
mercury-cadmium-telluride (MCT) detector. The application of polarization
modulation attenuates the noise of reflective FT-IR and the interference of water vapor
and carbon dioxide. The spectra given here are Reflectance-Absorbance (*RA*) given as:
$$RA = -\log(R/R_0) \qquad (2)$$
where $R$ and $R_0$ are the reflectance of fatty acid solution and pure water, respectively.
To obtain a better signal-to-noise ratio, spectra were collected with 2000 scans and 8
cm$^{-1}$ resolution at a fixed incidence angle of 40°. Data analysis was processed using
OPUS software for each displayed spectrum.
**2.5 Transmission electron microscope imaging**
Particle imaging was performed using a transmission electron microscope (TEM, FEI
Tecnai G2 F20, FEI, USA) equipped with a Schottky field emission gun. It was operated
at an accelerated voltage of 20–200 kV with a high angle annular dark field detector to
collect TEM images and even preserve the soft internal structure of organic sources
under high vacuum conditions.





**3 Results and discussion**

**3.1 SSA particle number size distributions**

To test the transfer of soluble saccharides and their interaction with insoluble fatty acids, experiments were carried out with seawater containing 1.0 g L$^{-1}$ glucose or trehalose, and particle number size distributions were obtained for each set of experiments. Figure 2 shows the particle number size distributions resulting from seawater to which different soluble saccharides were added in the presence or absence of fatty acids on the surface. As a reference, the particle size distribution produced from natural seawater is also given. The submicron particle size distributions produced by the plunging jet generator are well represented by lognormal mode. In the absence of saccharide, a broad, unimodal distribution of SSA with a peak number concentration around 168 nm was generated. This observation agrees quite well with previous studies that produced SSA by the plunging jet method (Christiansen et al., 2019; Prather et al., 2013). Moreover, the SSA yielded by plunging jet also has a size distribution similar to that yielded by the breaking wave, which particle number size distribution is ~162 nm. This contrasts with most previous laboratory studies using sintered glass filters or frits, which tend to exhibit a smaller mean diameter and narrower distribution. This may be because similar bubble size distributions exist in the two generation mechanisms using plunging jets and breaking waves. More notably, a marine accumulation mode with particle diameter of 170 nm (geometric mean diameter, 30% RH) was also employed in the widely used GEOS-Chem model based on the measurements of Quinn et al (Jaegle et al., 2011).



Laboratory studies of the effects of saccharide organic substances on droplet
production have been inconclusive. A previous study has used two bubble generation
methods (plunging water jet and diffusion aeration) to investigate the number size
distribution of SSA particles produced by mixing fructose and mannose with NaCl or
artificial seawater solution (King et al., 2012). The results showed that the yield of SSA
particles containing sodium dodecyl sulfate was significantly lower than that containing
fructose, but the yield of SSA particles containing mannose was lower than that
containing sodium laurate. Lv et al. (2020) found that addition fructose to sea salt
solution can significantly promote the increase of SSA number concentration. However,
the above studies lacked direct comparative results on SSA production influenced by
different soluble saccharides. For the plunging jet, our measurements indicate that
soluble saccharides can promote the production of SSA to varying degrees. It was
observed that glucose led to a slight increase in particle number concentration,
increasing the diameters to ~175 nm. As a contrast, the natural seawater spiked with
trehalose resulted in a higher total particle number concentration over a wide size range.
Therefore, the change in production and properties of SSA from actual seawater may
be more complicated under the influence of different saccharides.

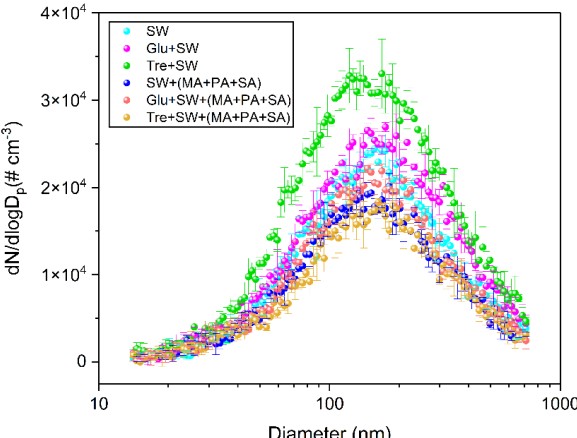


Figure 2. The particle number size distribution spectra of SSAs produced from blank

seawater sample and seawater sample spiked with glucose or trehalose. Both results are

presented here with and without fatty acid surface films.

The effect of the interaction of insoluble fatty acids with different saccharides on

SSA particles was investigated by spreading insoluble fatty acids on seawater surface.

In plain sight, fatty acids on the surface can significantly reduce the number

concentration of SSA regardless of the presence of saccharides in the seawater.

Moreover, fatty acids show the highest inhibitory effect on SSA produced by trehalose-

containing seawater solution. There are at least two mechanisms by which soluble

saccharides may affect particle production. On the first hand, it may stabilize or

destabilize bubbles on the water surface by associating with other substances favoring

co-aerosolization processes. On the other hand, soluble saccharides may influence

bubble bursting through changes in water and bubble surface tension. We ascribe that

the surface layer is significantly more stable, resulting in less bubble bursting in the

fatty acids case than in the glucose and trehalose case. Collectively, the observed



variability in these experiments suggests an urgent need to better build the link between
total SSA particle flux and seawater organic composition over the ocean. However, sole
bulk-phase generation experiments may not accurately capture the relevant chemical
behaviors and support mechanism analysis that occur in the SML. Therefore, we
attempted to explore the possible interaction mechanisms via air/water interface
chemical experiments.
**3.2 π-A isotherms of fatty acid monolayer**
In this section, we only discuss traditional Langmuir monolayers, which operate on
air/water interfaces that are ubiquitous along the sea surface (Elliott et al., 2014). The
π-A isotherm reflects information on the phase behavior of the monolayer as a function
of lipid packing density. As shown in Figure 3, the π-A isotherms of individual and
mixed fatty acids on the natural seawater subphase are presented. When the mechanical
barriers initially begin to compress, the amphiphilic molecules in the monolayer are in
the gaseous (G) phase under a large area per molecule, with the hydrophobic tails
having significant contact with the water surface, but little contact with each other. At
this stage, the compression of the film does not lead to a significant change in surface
pressure. As the monolayer is compressed, the intermolecular distances gradually
decrease and the surface pressure begins to rise from zero into the liquid expanded (LE)
phase, where the hydrophobic tails start to touch each other, but remain largely
disordered and fluid. This is represented as the lift-off area of the isotherm. Further
compression results in a thermodynamic transition to a liquid condensed (LC) phase.



The film is eventually compressed to a limiting point where the monolayer collapses as
the materials leaves the 2D film (Lee, 2008). In general, the collapse is an irreversible
process, and the collapsed material does not reintegrate into the monolayer as the
surface pressure decreases.
Although the π-A isotherms of individual fatty acids have been well studied, the
phase behavior of the mixed binary and ternary systems still needs to be further
explored. Pure natural seawater without spreading surface-active fatty acids does not
cause observable changes in the surface pressure, indicating that surface-active
impurities are either absent or have too low concentrations to cause film formation.
When myristic acid spreads on the water surface, it undergoes a long liquid phase, with
a lower collapse pressure of ~27 mN m$^{-1}$ and area per molecule as low as 5 Å$^2$. This is
due to the relatively high solubility of MA molecules in the aqueous phase, resulting in
a large loss of molecules in the monolayer. For palmitic acid monolayer, it goes through
a relatively short gaseous phase and rapidly enters the liquid phase. After experiencing
a kink point at ~40 mN m$^{-1}$, it continues to rise to ~52 mN m$^{-1}$ and collapses. Both the
lift-off area and molecular area of the stearic acid film decrease. This is caused by the
fact that the interaction (van der Waals force) between the molecules increases as the
molecular weight of long chain fatty acid increases. That is, increased attraction leads
to a decrease in distance between molecules.
When fatty acids are mixed in a certain molar ratio and spread onto the interface
water, it is found that the π-A isotherm lies between the pure fatty acids and is closer to
that of stearic acid, but the mean molecular area is relatively smaller. Consequently, the
longer fatty acids will dominate the lateral interactions of the SSA membrane, which
makes the membrane more rigid due to the larger sum of diffusive interactions. A
previous study has shown that PA and SA account for approximately two-thirds of the
total saturated fatty acids in fine SSA particles, with MA being the third most abundant
species (Cochran et al., 2016). In view of the true proportion of fatty acids in the nascent
sea spray particles, we used a ternary fatty acid membrane proxy system composed of
MA, PA, and SA (2:4:3 molar ratio) in the following experiments involving Langmuir
isotherms.

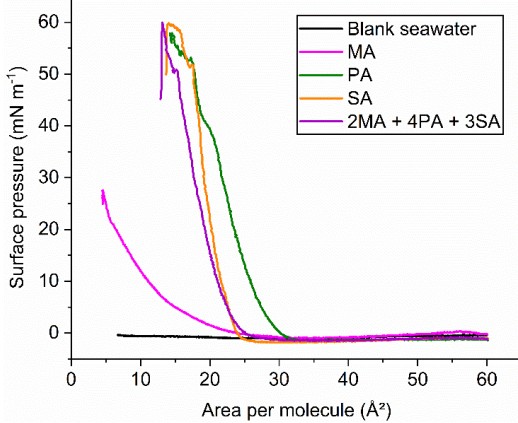


Figure 3. π-A isotherms of myristic acid, palmitic acid, stearic acid and mixed fatty
acids. The black trace represents the background natural seawater solution with no fatty
acid spread.

### 3.3 Effect of soluble saccharides on the phase behavior of mixed monolayers

An effective way to test whether soluble saccharides associate with lipid membranes is
to examine the effect of these saccharides on the phase behavior of lipid films. The π-
A isotherm provides us with rich information about the stability or fluidity of the



monolayer and molecular area under membrane compression (Nakata et al., 2012).
Both glucose and trehalose are highly soluble (>1.0 g L$^{-1}$) in water. However, this
solubility does not preclude their presence on the surface. According to some previous
studies, the dissolved organic carbon concentration is about 0.7–1.0 mg carbon L$^{-1}$
(Quinn et al., 2015; Hasenecz et al., 2019). Considering that saccharides in the ocean
represent approximately 20% of the dissolved organic carbon, the saccharide
concentration is about 0.14–0.20 mg L$^{-1}$ (De Vasquez et al., 2022). The Glucose and
Trehalose concentrations used for the π-A isotherms are approximately 3–4 orders of
magnitude greater than the saccharide concentration in dissolved organic matter,
maintaining detectivity within the π-A isotherms. Furthermore, high concentrations
used here are still relevant,considering the evaporation process in aged sea spray
aerosols (Hasenecz et al., 2020).
Figure 4 shows the π-A isotherms of mixed fatty acids on natural seawater subphases
containing different concentrations (varied between 0.1 and 5.0 g L$^{-1}$) of glucose or
trehalose. In this case, the surface pressure of the fatty acid monolayer is equal to that
of the fatty acid monolayer with the addition of saccharides, provided that the
saccharide molecules do not affect the monolayer. At a low concentration of 0.1 g L$^{-1}$,
both saccharides had little overall effect on the phase behavior of fatty acid monolayers.
However, they resulted in a smaller lift-off area for the monolayer compared to pure
natural seawater. As the Glucose and Trehalose subphase concentration increases, the
monolayers are expanded, taking up a larger mean molecular area, which is consistent
with previous research (Crowe et al., 1984). This noticeable expansion can be observed


from the lift-off to collapse, indicating that saccharides participate in and disrupt the
monolayer structure. This increase may result from Glucose and Trehalose displacing
a significant amount of surface water surrounding the lipid headgroups and integrating
into the mixed monolayer by forming hydrogen bonds, which leads to an increase in
the lateral area of the fatty acid molecules (Roy et al., 2016). Spectral evidence is
needed to further clarify whether intercalation occurs. Based on the results from π-A
isotherms, we conclude that the spacing of fatty acids in the monolayer by saccharides
also increases the fluidity of the membrane.
More surprisingly, we observed that the isotherms of the two saccharide matrices do
not exhibit much difference at the concentrations of 0.5 g L$^{-1}$ and 1.0 g L$^{-1}$. When the
saccharide concentration keeps increasing to 5.0 g L$^{-1}$, the molecular packing density
on the interface decreases, and the apparent molecular area increases. In the presence
of glucose and trehalose, the lift-off areas increased by 9 and 10 Å$^2$, respectively. Thus,
it is effectively demonstrated that the more soluble saccharide molecules are added, the
more fatty acid molecules joined to more saccharides or more sites per saccharide
molecule remain separated. Another distinguishing feature of the fatty acid isotherms
is the change of slope above ~40 mN m$^{-1}$. This result could be interpreted as the
saccharide being "squeezed" out of the insoluble film, resulting in higher monolayer
compressibility. By squeezing saccharide molecules out of the monolayer, the isotherms
at high surface pressure behave similarly to other isotherms with low saccharide
concentrations. The difference is that with the increase of structural complexity of
saccharide, the effect of Trehalose at the same concentration is more prominent. The α,
α, 1,1-linkage between two glucose subunits in trehalose is considered to provide an
elastic and rigid balance, thus allowing for strong interactions with multiple fatty acid
headgroups (Clark et al., 2015). The expansion effect promoted by soluble saccharides
is more relevant at lower surface pressure when alkyl chains are farther apart from each
other.

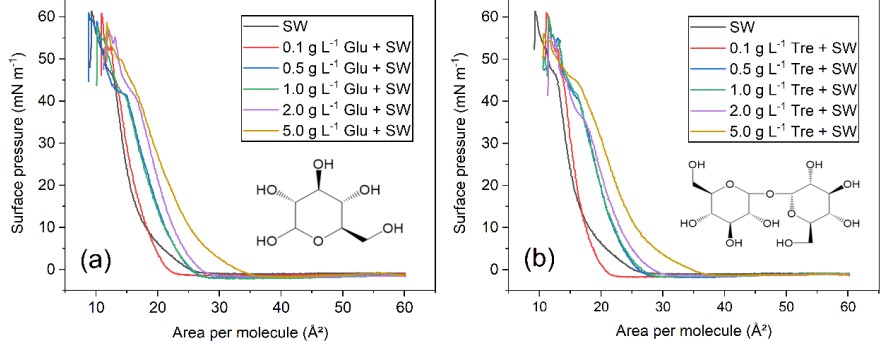


Figure 4. π-A isotherms of mixed fatty acids in the SW subphase with several
concentration gradients of (a) glucose, and (b) trehalose. The inset shows the molecular
structures of glucose and trehalose.
The existence of such expansion behavior in the presence of saccharides implies a
degree of complexity and heterogeneous distribution of species in the interfacial region.
In addition, trehalose exhibits more affinity for fatty acids in the monolayer than
glucose, in part because trehalose interacts less with neighboring saccharide molecules
(Leekumjorn and Sum, 2008). These results suggest that trehalose tends to bind to
monolayer surfaces better than glucose, forming a subsurface. This difference can be
explained by the fact that glucose is smaller than trehalose and has greater mobility in
combination with other glucose molecules. As a result, trehalose binds more readily to
lipid monolayer surfaces than glucose and is less mobile, as is evident from
experimental observations. This is consistent with the result of Crowe et al. on the effect
of saccharides (glucose, sucrose, trehalose and raffinose) on the properties of 1,2-
dimyristoyl-*sn*-glycero-3-phosphocholine (DMPC) and 1,2-dipalmitoyl-*sn*-glycero-3-
phosphocholine (DPPC) monolayers. That is, the area per lipid increases with the
increase of saccharide concentration, and trehalose provides the largest lateral
monolayer expansion (Crowe et al., 1984). Clarifying and refining the interaction
mechanisms by which lipid molecules interact with saccharides is critical to any attempt
to model such chemical phenomena occurring at environmentally relevant interfaces.
**3.4 Effect of soluble saccharides on the interfacial structure of mixed monolayers**
PM-IRRAS is a surface sensitive technique that allows further study of the possible
effects of soluble saccharides on lipid interfacial organization at the molecular level.
Figure 5 shows the IRRAS spectra for mixed fatty acid monolayers at two different
saccharides containing subphases at a surface pressure of ~30 mN m$^{-1}$. This phase
corresponds to the two-dimensional LC phase. Figure S3 shows the IRRAS spectra of
mixed fatty acids measured at different surface pressures. It can be observed that with
the increase of surface pressure, the intensity of the peaks also increases accordingly,
reaching a relatively stable level around 30 mN m$^{-1}$. Considering the stability of the
monolayer, this surface pressure was chosen to obtain the desired infrared spectra.
The absorption band in the 3000–2800 cm$^{-1}$ region shown in Figure 5 is ascribed to
the CH stretching vibration of the alkyl chain. The main features at ~2916 and ~2850
cm$^{-1}$ are related to antisymmetric ($\nu_{as}(CH_2)$) and symmetric ($\nu_s(CH_2)$) stretching modes



of methylene of mixed fatty acids, respectively. The $\nu_{as}(CH_2)$ feature consistently
remains stronger than $\nu_s(CH_2)$ with the increase of Glucose and Trehalose
concentrations. These two band positions are often used to be empirically correlated
with the order and organization within the alkyl monolayer adsorbed to the water
interface, with higher wavenumbers corresponding to disordered *gauche* conformers.
Conversely, low wavenumbers indicate that the alkyl chain of lipids is well ordered
with preferential *all-trans* characteristics. In this work, the relatively low frequencies
of $\nu_{as}(CH_2)$ (2916 cm$^{-1}$) and $\nu_s(CH_2)$ (2850 cm$^{-1}$) hint that the molecular conformation
of the fatty acid alkyl chains is dominated by the highly ordered *all-trans* conformation
(Li et al., 2019). Despite the concentration range of saccharides varied widely, the
positions of $\nu_{as}(CH_2)$ and $\nu_s(CH_2)$ showed modest sensitivity to shift, suggesting very
minor changes in the conformation of the alkyl chain. The relative weak antisymmetric
($\nu_{as}(CH_3)$) and symmetric methyl stretching ($\nu_s(CH_3)$) vibrations were observed at
~2958 and ~2877 cm$^{-1}$, respectively. These results indicate that the penetration of
soluble saccharides is only superficial (along the lipid surface) and has little effect on
the alkyl tail arrangement. Therefore, it is further deduced that the stabilization
mechanism between saccharides and fatty acid molecules may occur in the headgroup
region.

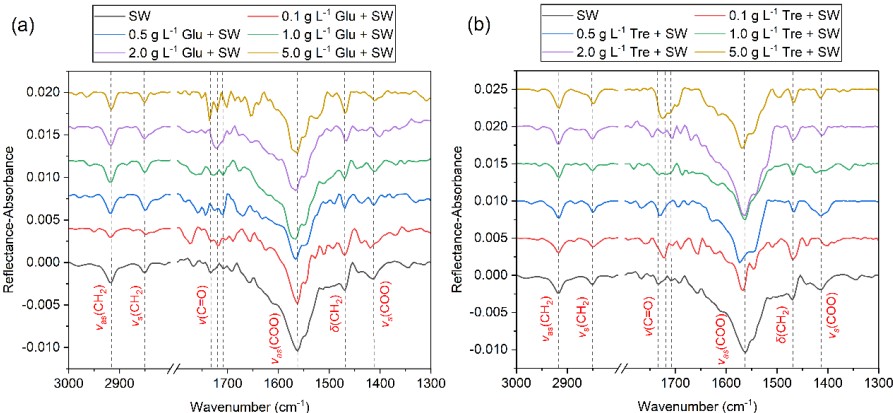

Figure 5. PM-IRRAS spectra of mixed fatty acids at the air/seawater interface at different (a) glucose, and (b) trehalose concentration in the subphase.

Carboxylic acids possess one hydrogen bond donor (hydroxyl) and one hydrogen bond acceptor (carbonyl) within the same functional group, the carboxyl group. The carbonyl stretch mode ($v$(C=O)) of the carboxyl group at ~1732 cm$^{-1}$ (unhydrogen bonded) was observed in seawater. This band component is put down to the conformation with the carbonyl group almost parallel to the water surface. In the presence of saccharides, the unhydrated C=O band was observed to be depressed, and the singly and doubly hydrogen bonded carbonyl components at ~1720 and ~1708 cm$^{-1}$ became dominant (Johann et al., 2001). The presence of hydrogen bonds between saccharides and the carbonyls of fatty acids is well correlated with the observed shifts in the infrared absorption band of carbonyl groups. Using FTIR experiments, Luzardo et al. (2000) showed that trehalose shifts the vibrational frequency of the carbonyl group to a lower value, which is an evidence of the existence of direct hydrogen bonding between trehalose and lipid carbonyl groups. We believe that saccharides displace water surrounding the fatty acid polar headgroups and interact strongly with both water and





lipid headgroups, resulting in a slight increase in hydration near the monolayer interface.

468       The nonmonotonic hydrogen bond strength shows that the interaction at the interface

manifests as competing contributions that dominate at different concentrations. Within
the concentration range studied, saccharides tend to "displace" water, creating unique
environments. In some recent studies, this "water displacement" hypothesis was
supported by molecular dynamics (MD) simulations, fluorescence microscopy and
nuclear magnetic resonance (NMR) (Lambruschini et al., 2000; You et al., 2021; Kapla
et al., 2015). Previous MD simulation studies showed that the hydrogen bond lifetime
between trehalose and membrane was longer than that established between water and
membrane (Villarreal et al., 2004). This is because water molecules are more mobile
and can exchange more frequently at the interface than trehalose. Another study also
confirmed that sugar-lipid hydrogen bonds are stronger than water-lipid hydrogen
bonds due to low endothermicity and they remain largely intact even at very high sugar
concentrations (You et al., 2021).

481       Long chain fatty acid amphiphiles that spread as a monolayer on the alkaline

subphase undergo dissociation. The ratio of neutral fatty acids and ionized carboxylates
in the monolayer depends on the pH of the subphase solution. At natural oceanic
conditions (pH~8.0), deprotonation of the carboxylic acid groups results in two
carboxylate stretches. The broad and strong antisymmetric carboxylate stretch
($v_{as}$(COO)) were observed at ~1562 cm$^{-1}$, and the symmetric carboxylate stretch
($v_{s}$(COO)) at ~1412 cm$^{-1}$. The presence of salt in seawater caused the $v_{as}$(COO) to split
into three peaks at ~1562, ~1547 and ~1524 cm$^{-1}$. A distinctive feature in all spectra





obtained at ~1469 cm$^{-1}$ was assigned to the CH$_2$ scissoring vibration ($\delta$(CH$_2$)) of the
aliphatic chain (Muro et al., 2010). This wavenumber value somewhat indicates an
orthorhombic subcell structure. It should be noted that the $\delta$(CH$_2$) vibrational position
for the surface membrane of the mixed fatty acids reported here is relatively insensitive
to saccharides and their concentrations. This observation confirms the conclusions
drawn from the $\nu_{as}$(CH$_2$) and $\nu_s$(CH$_2$) wavenumbers that higher alkyl chain
conformational orders are obtained either on the surface of pure seawater or on
subphases containing Glucose or Trehalose.

**3.5 Effect of soluble saccharides on particle morphology**

Particle morphology can affect the surface composition, heterogeneous chemistry, gas-
particle partitioning of semi-volatile organics and water uptake of aerosols (Unger et
al., 2020; Ruehl et al., 2016; Lee et al., 2021). We examined the particle morphology
and qualitatively compared SSAs between different model systems, including the
mixed effects of saccharides and fatty acids. Compared to the study by Unger et al.
(2020), the samples we investigated had compositions that were closely connected to
the chemical composition of sea spray aerosols.

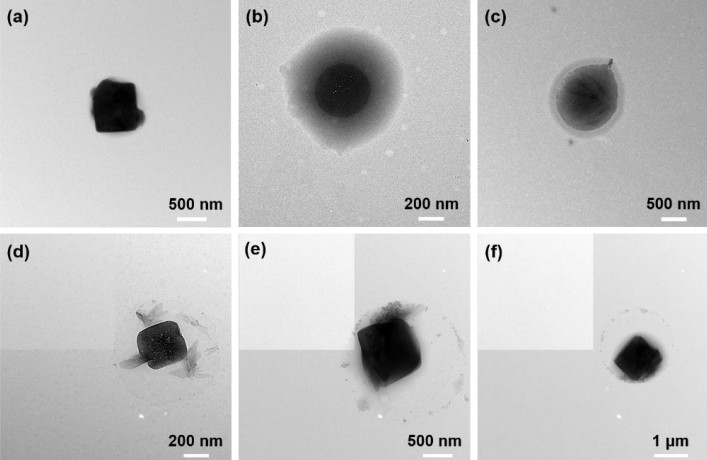


Figure 6. TEM images of morphology identified for sea spray aerosols produced from
(a) natural seawater, (b) seawater with glucose and (c) seawater with trehalose without
fatty acids organic layer; (d) natural seawater with fatty acids, (e) seawater with glucose
and fatty acids, (f) seawater with trehalose and fatty acids.
Figure 6 depicts TEM images of SSA particles generated by plunging jet sea spray
aerosol generator, which can provide clues about how saccharide and/or fatty acid
components are interacting with sea salt. As can be seen from the Figure 6a, SSA
produced from pure natural seawater by plunging jet exhibited a prism-like morphology
that is predominantly inorganic in nature (Lee et al., 2020). This standard cubic shape
also suggests that NaCl is an important component of natural seawater sample used in
this study. The morphology of SSA particles was strongly affected by the incorporation
of saccharides. In the presence of saccharides, the images indicate that these SSA
particles exhibit a core-shell morphology with the shell portion being mainly organic
in composition, whereas sea salt core are more spherical in nature, demonstrating that
organic substances inhibit the cubic crystallization of NaCl. The core-shell



morphologies adopted here are congruent with previous studies on the NaCl/Glucose
binary system and authentic SSA samples observed using atomic force microscopy
(Ray et al., 2019; Estillore et al., 2017).
The particles in Figures 6b–c existing as mixtures of salt and organic matter are also
commonly labeled as sea salt-organic carbon (SS-OC) (Ault et al., 2013a). The core-
shell morphology was found to be highly dependent on the salt-to-organic ratio and
varied with the nature and solubility of organic components (Estillore et al., 2017). For
example, previous studies have showed that adding organic material to aqueous
solution, particle morphology, crystallization behavior and optical properties were
changed as the amount of organic content in the particles increased (Ault et al., 2013b;
Freedman et al., 2009). Furthermore, the shell thickness of core-shell SSA shows size-
dependent variability. Specifically, larger core-shell particles generally displayed
relatively thinner coatings, while smaller core-shell particles displayed thicker coatings.
However, as shown in Figures 6d–f, the presence of fatty acid layer on the surface not
only has little effect on the morphology of SSA particles, but also weakens the
morphology modification of SSA particles by saccharides. In a word, the results
presented in this study suggest that the heterogeneity within a particle type is a function
of seawater chemistry.
**3.6 Proposed mechanism for bulk saccharide transfer to SSA**
The molecular level interactions between small saccharides and fatty acids discussed in
previous sections can be summarized using the model presented in Figure 7. Aqueous

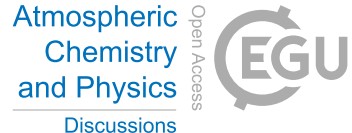

aerosols coated by surface-active organic matter (Figure 7a), such as SSA, generally
hold inverse micelle structures with hydrophilic headgroups pointing toward the
aqueous phase and hydrophobic tails pointing toward the gas phase (Blackshaw et al.,
2019). At the center of the inverse micelle, a water pool is formed that can dissolve
polar substances such as saccharides, proteins, enzymes, amino acids and nucleic acid.
This unique physicochemical environment may enhance the possibility of saccharides
transfer to SSA. Through the Langmuir surface pressure-area experiment combined
with infrared reflection-absorption spectroscopy, we initially explored the possible
mechanism of the transfer of saccharides at the air/water interface. In a nutshell, we
infer that saccharides initially in the aqueous phase move steadily to the interface and
act as a substituent for water molecules, and locate in the headgroup region of the fatty
acids. During the binding process, the saccharides displace the oriented water molecules
that are bound to the fatty acids through hydrogen bonds, establishing new hydrogen
bonds with the carbonyl group of fatty acids (You et al., 2021).

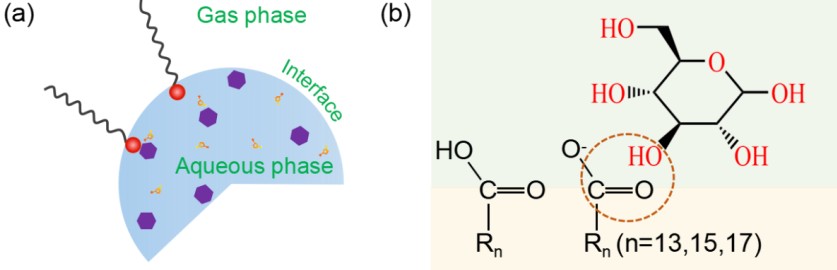


Figure 7. (a) Proposed model of fatty acid-saccharide interaction at the air/water
interface. (b) Description of possible mechanisms of fatty acid-saccharide interaction
at the air/water interface.





### 3.7 Atmospheric implications

Despite extensive efforts, the exhaustive relationships between ocean organic carbon pools and the chemical composition of SSAs are still outstanding. The coupling of this sea spray aerosol simulation generator with the interfacial monolayer model lays the foundation for further studies of the material relationship between the ocean and SSA. The research reported here yielded two key findings. First, the SSA production and particle size distribution are usually extremely sensitive to organic matter, and small saccharides dissolved in seawater are critical to the formation, size and composition of SSA. Our results strongly support that saccharides can greatly promote the generation of SSA particles and make SSA show core-shell morphology characteristics. A previous study revealed that the SSA number concentration in coastal samples was inversely correlated with salinity, with several organic tracers, including dissolved and chromophoric organic carbon (DOC, CDOM), marine microgels, and chlorophyll a (Chl-a) being positively correlated, but not associated with viral and bacterial abundances (Park et al., 2019). Therefore, the factors that affect the emission of SSA have not been fully clarified. Other limitations to this study include the poor representation, by the simple chemical structural models, of the myriad complex biomolecules that exist in the ocean, spanning dissolved, colloidal and particulate matter. It is recommended that future studies targeting the production and property of SSA include the effects of different types of organic matter to determine whether they fully mimic the arrays of SSA particles, and include more complete organic matter systems as well as biological species.





Second, it has been suggested that the abundant organic content in SSA plays a key
role in determining the cloud condensation nucleation and ice nucleating activity
(O'dowd et al., 2004). Therefore, climate models demand a predictive representation of
SSA chemical composition to accurately simulate climate processes in the marine
boundary layer (Burrows et al., 2016; Bertram et al., 2018). However, the source of
organic enrichment observed in SSA remains speculative, which poses challenges to
the modeling of the aerosol impact on atmospheric chemistry and climate science. A
recent study has raised that the cooperative adsorption of saccharides with insoluble
lipid monolayers may make important contributions to the sea spray aerosols and even
have climatic consequences with broad research prospects (Burrows et al., 2014). In
this work, we used the Langmuir monolayer to model possible interactions between
subphase soluble saccharides and surface fatty acid molecules. A subsequent study used
infrared reflection-absorption spectroscopy to determine the interaction mechanism
between two simple soluble saccharides and tightly packed fatty acids monolayers at
the air/water interface. Combining the above experimental data, we infer that a
hydrogen bond network between saccharides and the carbonyl group of surface
insoluble fatty acid molecules contributes to its transfer from the ocean to the
atmosphere. At present, this mechanism of saccharide transfer and enrichment has not
been emphasized in the model describing SSA formation. To further examine the
feasibility of the hydrogen bonding mechanism as an interfacial organic enrichment
mechanism, it is necessary to further explore and verify the interaction of other
carbohydrates with common surface-insoluble molecules in future studies.





**4 Conclusions**

In summary, we simulated the production of SSA in natural seawater spiked with two common soluble saccharides using a plunging water jet generator and revealed the possible mechanism of saccharide transfer from bulk seawater into SSA combined with surface sensitive infrared spectroscopy techniques. We confirmed that glucose and trehalose can significantly promote the production of SSA and alter the surface morphology of SSA particles. This highlights the potential for a direct oceanic source of carbohydrate organics through bubble bursting. In addition, trehalose showed stronger promoting ability than glucose, while the surface fatty acid layer played an inhibitory role. Using the mixture of saturated fatty acids MA, PA and SA as the proxy of SSA surface film, the π-A isotherms provided strong evidence that saccharides can interact with insoluble fatty acid monolayers and be absorbed at the monolayer, which caused expansion of the monolayer and made the films heterogeneous. According to the IRRAS spectra, soluble saccharides did not produce an observable effect on the order of fatty acid alkyl chains. We further infer that soluble saccharides are mainly located on the subsurface below the monolayer, and interact with carbonyl groups of fatty acids by forming hydrogen bonds to facilitate their sea-air transfer. Crucially, this work provides physical and molecular signatures of potentially important saccharides transfer mechanism with general implications for understanding how saccharide-lipid-water interactions affect sea spray aerosol systems.



**Data availability**

Data are available by contacting the corresponding author.

**Supplement**

The supplement related to this article is available online at:

**Author contributions**

MX: conceived the experiment, data curation, formal analysis, writing original draft, writing-review & editing. NTT: writing - review & editing. JL: writing - review & editing. LD: supervision, conceived the experiment, funding acquisition, writing - review & editing.

**Competing interests**

The author declare that they have no conflict of interest.

**Financial support**

The authors acknowledge support from National Natural Science Foundation of China (22076099, 21876098), the Department of Education of Shandong Province (2019KJD007), and Fundamental Research Fund of Shandong University (2020QNQT012).



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
