# Peer review of "Insoluble lipid film mediates transfer of soluble saccharides from the sea to the"

_Atmospheric Chemistry and Physics, 2022_

## Author Comment (AC1)

**Authors' Response to Reviews of**

"Insoluble lipid film mediates the transfer of soluble saccharides from the sea to the atmosphere: the role of hydrogen bonding"

Minglan Xu, Narcisse Tsona Tchinda, Jianlong Li, Lin Du*

We thank the Referee for the constructive comments. We have addressed the comments point by point below and modified the manuscript accordingly. For clarity, the Referee's comments (RC) are reproduced in blue, authors' responses (AR) are in black and changes in the manuscript are in red color text.

**Anonymous Referee #1**

This manuscript presents a comprehensive study on the transfer of soluble saccharides at the sea-air interface, which affects the number concentration, chemical composition, and morphology of the resulting sea spray aerosols. I appreciate the fascinating experimental methodology of the study, and I enjoy digesting the results. The combination of Langmuir monolayer technology and infrared reflection-absorption spectroscopy helps to explain the interaction between soluble saccharides and insoluble fatty acids, and thus deduce a unique mechanism of hydrogen bonding, which is a novel technique and interesting results presentation, and certainly deserves more attention. Overall, the manuscript is well-written and most important the authors do a brilliant job in the reference list, very multidisciplinary and very updated. The authors have considered multiple views, and it is worth considering to be published in ACP after addressing some general issues.

RC: I have two questions about the abstract. Whether insoluble lipid monolayers can be absorbed at the air/water interface or simply float on the surface? Regarding hydrogen bond analysis, I think the authors are more illustrated by the changes in the infrared spectra of the carbonyl region, and it is necessary to consider whether the expression here is appropriate.

AR: To be a surfactant, a molecule must consist of distinct hydrophobic and hydrophilic portions. A common example is a fatty acid comprising a long hydrocarbon chain attached to a -COOH head group. The hydrophilic head group anchors in the water surface whilst the hydrophobic tail prefers to extend out of the water into the water/air interface. The term 'insoluble' refers to surfactants that, when placed at the water/air interface, will preferably remain there instead of dissolving in the bulk water. Therefore, this surfactant can be adsorbed at the water/air interface and studied uniquely because their concentration at the interface is known directly, from the amount added to the interface. We have modified the sentence at page 2 as:

Langmuir model was used to parameterize the adsorption and distribution of saccharide into SSA across the bubble surface, while infrared reflection-absorption spectroscopy (IRRAS) combined with Langmuir isotherms were undertaken to examine the effects of aqueous subphase soluble saccharides on the phase behavior, structure and ordering of insoluble lipid monolayers adsorbed at the air/water interface.

AR: Indeed, the interaction mechanism between fatty acids and soluble saccharides is mainly inferred by the changes of characteristic bands in the carbonyl region. We have changed the expression in the revised manuscript. We have rewritten this sentence at page 2-3:

The enhancement of the carbonyl band to the low wavenumber region implied that soluble saccharides can form new hydrogen bonds with fatty acid molecules by displacing large amounts of water near the polar head groups of fatty acids.

RC: Line 104: "A possible explanation for the SSA composition in saccharides involves", whether or not the original meaning of expression should be the saccharides in the SSA composition, perhaps this sentence needs to be rewritten.

AR: Originally, we intended to express that a possible explanation for the origin of saccharides in SSA composition involves the affinity between soluble saccharides and insoluble surfactant monolayers adsorbed at the water/air interface, leading to co-adsorption of saccharides. The sentence has been rewritten in the revised manuscript at page 6:

A possible explanation for the origin of saccharides in SSA chemical composition involves the affinity between the bulk aqueous soluble saccharides and insoluble surfactant monolayers already adsorbed at the air/water interface, resulting in co-adsorption of the soluble saccharides (Link et al., 2019b).

RC: Line 148-149: It is best for the authors to explain why this molar ratio is used to obtain the mixed fatty acid solution in the method section, rather than in the results and discussion section below.

AR: We fully agree with the Referee that it might seem more appropriate to adjust the specific interpretation of the mixing molar ratio to the method section. Therefore, in the method section at page 8, we have added a text that explains why the mixed fatty acid solution is obtained in the indicated molar ratio:

The respective fatty acid solutions were mixed at a molar ratio of 2 MA:4 PA:3 SA to obtain a mixed lipid stock solution considering that PA and SA account for approximately two-thirds of the total saturated fatty acids in fine SSA particles, with MA being the third most abundant species (Cochran et al., 2016).

RC: In the experiment of preparing Langmuir monolayer, whether the volatilization time of 15 min is enough to make chloroform volatilize completely?

AR: Chloroform is a common spreading solvent used in Langmuir monolayer experiments (Aoki et al., 2016; Adams et al., 2016; Carter-Fenk and Allen, 2018). It can dissolve most organic compounds and can quickly evaporate when its solution is spread dropwise onto the aqueous surface. Previously, we have used stearic acid as an example to study the effect of solvent evaporation time on its π-A isotherm. Here, 5 min, 10 min, 15 min, 20 min and 30 min were selected respectively, and it was found that when waiting for 15 min, 20 min and 30 min, the π-A isotherms obtained were very close to each other, which could indicate that the volatilization time of 15 min was enough to make the solvent volatilize completely, and also proved that prolonging the waiting time would not lead to the dissolution of stearic acid molecules in the liquid phase. In addition, in the case of volatilization for 5 and 10 min, it was found that the π-A isotherms moved to the direction of larger molecular area, indicating that the solvent may not be volatilized completely.

[Figure]

Fig R1. π-A isotherms of stearic acid on pure water subphase at different volatilization times.

RC: Line 226: Whether R and $R_0$ in the eq2 should be the surface covered by fatty acid monolayers and the surface of pure seawater solution?

AR: The spectra presented here are reflectance-absorbance (RA), where RA=-log(R/$R_0$) and R and $R_0$ are the reflectivity of the analyte (fatty acid) and pure aqueous solution (natural seawater), respectively. We have explained this in more detail in the revised manuscript at page 12:

The spectra presented here are reflectance-absorbance (*RA*) given as:

$$RA = -\log (R/R_0) \qquad (2)$$

where $R$ and $R_0$ are the reflectance of fatty acid monolayer and pure seawater solution surface, respectively.

RC: Section 3.1 The authors have compared the SSA produced using the sea spray aerosol generator in this study with other laboratory devices, and obtained similar consistency, but lack a comparison with the particle number size distribution of the real SSA observed in the field. In other words, whether the device can fully simulate the real SSAs?

AR: We compared the particle number size distribution of SSAs generated using the plunging-jet sea spray aerosol generator in this study with the real SSAs measured in the field observations (Xu et al., 2022; Quinn et al., 2017), as shown in Figure R2. In the submicron size range, it was observed that the size distribution of both laboratory-generated SSAs and SSAs measured in the field had a major accumulation mode in the range of 111-172 nm. The number concentration of SSAs generated in the laboratory is about 2 orders of magnitude higher than observed in the real environment. As a result, the jet sea spray generator system is capable of a wide range of measurements (e.g., size-resolved hygroscopicity and heterogeneous reactivity) that are not achievable at low number concentrations. In addition, the experiments are repeatable and provide many benefits for the study of the chemical and physical control of marine bubbles, foams and aerosols. We have supplemented the comparison with the field SSAs in the revised manuscript at page 13-14:

[Figure]

Fig R2. Comparison of particle number size distribution of SSAs generated by sea spray aerosol generator in our study with real SSA in the field measurements.

Moreover, we compared the particle size distributions of SSA generated in our laboratory with those measured in field studies (Quinn et al., 2017; Xu et al., 2022). As shown in Figure S3, it was observed that the size distribution of both

laboratory-generated SSAs and SSAs measured in the field had a major accumulation mode in the range of ~111–172 nm. However, the number concentration of SSAs produced in our experiment is about 2 orders of magnitude higher than that in the real environment. As a result, the jet sea spray generator system is capable of a wide range of measurements (e.g., size-resolved hygroscopicity and heterogeneous reactivity) that are not achievable at low number concentrations.

RC: A drawback of the presented results is the lack of quantitative information describing the SSA production, specifically the variation of SSA number concentration. A clearer presentation of measured aerosol number concentrations would be most helpful.

AR: Thank you for the very helpful comments. We agree that the results may be difficult to follow without quantitative analysis. Therefore, we have made corresponding quantitative supplements in the revised manuscript at page 15-16. For more clarity, we also added detailed information of the number concentration and mass concentration obtained by each group of experiments in the supplement (Table S1).

It was observed that glucose led to a slight increase of about 15.6% in particle number concentration, increasing the mode diameters to ~175 nm. In contrast, the natural seawater spiked with trehalose resulted in a higher total particle number concentration that increased by approximately 49.4% over a wide size range.

When the fatty acid surfactant was added to seawater alone, the number concentration decreased by about 17.2%, while the presence of glucose resulted in a decrease of about 21.6%. Moreover, fatty acids showed the highest inhibitory effect on SSA produced by trehalose-containing seawater solution, whose concentration decreased by about 49.4%.

RC: Line 261-264: An ambiguous sentence. According to the above description, the author here should be referring to the comparison of SSA particles produced by artificial seawater solution containing such organic matter, rather than SSA particles containing these substances.

AR: We have rewritten this sentence in the revised manuscript at page 14:

The results showed that the number concentration of particles produced by artificial seawater containing sodium dodecyl sulfate was significantly lower than that of particles produced by artificial seawater containing fructose. However, NaCl solution containing mannose produced SSA with lower number concentration than NaCl solution containing sodium laurate.

RC: Line 325-326: Here the decrease in the lift-off area and molecular area of the stearic acid film should be relative to the previous palmitic acid. Therefore, this sentence should be correct only if it is added with respect to palmitic acid.

AR: We have modified this sentence in the revised manuscript at page 18:

> Both the lift-off area and molecular area of the stearic acid film decreased more than those of palmitic acid film.

RC: Line 355-360: In the Langmuir isotherms and infrared spectroscopy experiments, the authors designed concentrations about 3 orders of magnitude higher than in the real environment, and said that this high concentration is still environmentally relevant, how can it be better explained?

AR: On the one hand, the concentration of glucose and trehalose chosen in the experiment about 3 orders of magnitude higher than the marine environment is to maintain the detectability of the $\pi$-A isotherms. In addition, high concentrations remain relevant due to the evaporation process in aged sea spray aerosols (Hasenecz et al., 2020; Hasenecz et al., 2019). On the other hand, such high concentrations are close enough to remain relevant in the understanding of saccharide enrichment in the sea surface microlayer while facilitating confident interpretation of the physicochemical mechanism driving the adsorption and transfer of soluble saccharides. We have provided appropriate supplementary explanations in the revised manuscript at page 19:

> At the same time, such concentrations are close enough to understand the enrichment of saccharides in sea surface microlayer and to provide a confident interpretation of the physicochemical mechanisms driving the adsorption and transfer of soluble saccharides (De Vasquez et al., 2022).

**Some minor comments:**

RC: Line 43: It is better to use the generic name: humic-like substances (HULIS).

AR: This was changed in the revised manuscript.

RC: Line 157: "solution" should be changed to plural "solutions".

AR: This has been corrected in the revised manuscript.

RC: Line 272: "change"-"changes"

AR: This was changed in the revised manuscript.

RC: Line 355: There are several instances in the manuscript where "Glucose and Trehalose" should be lowercase.

AR: This was changed in the revised manuscript.

**References**

Adams, E. M., Verreault, D., Jayarathne, T., Cochran, R. E., Stone, E. A., and Allen, H. C.: Surface organization of a DPPC monolayer on concentrated $SrCl_2$ and $ZnCl_2$ solutions, Phys. Chem. Chem. Phys., 18, 32345-32357, 10.1039/c6cp06887a, 2016.

Aoki, P. H. B., Morato, L. F. C., Pavinatto, F. J., Nobre, T. M., Constantino, C. J. L., and Oliveira, O. N.: Molecular-level modifications induced by photo-oxidation of lipid monolayers interacting with erythrosin, Langmuir, 32, 3766-3773, 10.1021/acs.langmuir.6b00693, 2016.

Carter-Fenk, K. A. and Allen, H. C.: Collapse mechanisms of nascent and aged sea spray aerosol proxy films, Atmosphere, 9, 503, ARTN 503 10.3390/atmos9120503, 2018.

Hasenecz, E. S., Kaluarachchi, C. P., Lee, H. D., Tivanski, A. V., and Stone, E. A.: Saccharide transfer to sea spray aerosol enhanced by surface activity, calcium, and protein interactions, ACS Earth Space Chem., 3, 2539-2548, 10.1021/acsearthspacechem.9b00197, 2019.

Hasenecz, E. S., Jayarathne, T., Pendergraft, M. A., Santander, M. V., Mayer, K. J., Sauer, J., Lee, C., Gibson, W. S., Kruse, S. M., Malfatti, F., Prather, K. A., and Stone, E. A.: Marine bacteria affect saccharide enrichment in sea spray aerosol during a phytoplankton bloom, ACS Earth Space Chem., 4, 1638-1649, 10.1021/acsearthspacechem.0c00167, 2020.

Quinn, P. K., Coffman, D. J., Johnson, J. E., Upchurch, L. M., and Bates, T. S.: Small fraction of marine cloud condensation nuclei made up of sea spray aerosol, Nat. Geosci., 10, 674-679, 10.1038/ngeo3003, 2017.

Xu, W., Ovadnevaite, J., Fossum, K. N., Lin, C. S., Huang, R. J., Ceburnis, D., and O'Dowd, C.: Sea spray as an obscured source for marine cloud nuclei, Nat. Geosci., 15, 282-286, 10.1038/s41561-022-00917-2, 2022.

---

## Author Comment (AC2)

**Authors' Response to Reviews of**

"Insoluble lipid film mediates the transfer of soluble saccharides from the sea to the atmosphere: the role of hydrogen bonding"

Minglan Xu, Narcisse Tsona Tchinda, Jianlong Li, Lin Du\*

We thank the Referee for the constructive comments. We have addressed the comments point by point below and modified the manuscript accordingly. For clarity, the Referee's comments (RC) are reproduced in blue, authors' responses (AR) are in black and changes in the manuscript are in red color text.

**Referee #2**

This manuscript presents an important contribution to the understanding of saccharide transfer and enrichment in sea spray aerosol. The authors conduct a thorough experimental investigation in which both particle properties and fundamental physicochemical properties of the air/water interface are investigated. The molecules used within this study represent important and abundant contributors to sea spray aerosol composition, and the composition was carefully selected to serve as a good model system. Additionally, the manuscript is well-written, and the results are summarized clearly and succinctly. While I think that the main arguments presented in the manuscript are reasonable, I believe that several clarifying details and/or a few additional control experiments will make the conclusions more convincing.

RC: What was the pH of the seawater solution at the beginning and end of the experiments? Basic solutions acidify over time when exposed to air due to atmospheric carbon dioxide, and pH changes can dramatically change the film morphology and saccharide adsorption to the monolayer. Carter-Fenk and Allen (Carter-Fenk, K. A.; Allen, H. C. Collapse Mechanisms of Nascent and Aged Sea Spray Aerosol Proxy Films. Atmosphere 2018, 9 (12), 503. https://doi.org/10.3390/atmos9120503.) demonstrate these pH-dependent changes using the same proxy monolayer mixture, 2 MA : 4 PA : 3 SA. The film morphology and isotherm change as a function of pH, and the myristic acid solubility decreases with decreasing pH. Consequently, any solution acidification could enhance myristic acid adsorption to the air/water interface, thereby expanding the monolayer and increasing the observed mean molecular area in the surface pressure-area isotherms. Carter-Fenk et al. also show how the subphase pH impacts saccharide co-adsorption to a palmitic acid and cetyl alcohol monolayer, albeit using different saccharides (Carter-Fenk, K. A.; Dommer, A. C.; Fiamingo, M. E.; Kim, J.; Amaro, R.; Allen, H. C. Calcium Bridging Drives Polysaccharide Co-Adsorption to a Proxy Sea Surface Microlayer. Phys. Chem. Chem. Phys. 2021, 23 (30), 16401–16416. https://doi.org/10.1039/D1CP01407B). The fatty acid carboxylic acid protonation

state can change near seawater pH, and the overall monolayer protonation state impacts the intermolecular interactions between saccharides and monolayer headgroups at the air/water interface. For further discussion on the surface pKa of fatty acids at the air/water interface, see the following references: Wellen, B. A.; Lach, E. A.; Allen, H. C. Surface pKa of Octanoic, Nonanoic, and Decanoic Fatty Acids at the Air–Water Interface: Applications to Atmospheric Aerosol Chemistry. Phys. Chem. Chem. Phys. **2017**, 19 (39), 26551–26558. https://doi.org/10.1039/C7CP04527A.; Zhang, T.; Brantley, S. L.; Verreault, D.; Dhankani, R.; Corcelli, S. A.; Allen, H. C. Effect of pH and Salt on Surface pKa of Phosphatidic Acid Monolayers. Langmuir **2018**, 34 (1), 530–539.

https://doi.org/10.1021/acs.langmuir.7b03579.

AR: As the Referee said, the strong change in pH from basic to neutral and acidic does have a great effect on the film morphology and  $\pi$ -A isotherm. The proxy monolayer mixture (2 MA: 4 PA: 3 SA) was spread on NaCl subphase at pH 8.2, 5.6 and 2.0 to simulate the composition of SSA aqueous cores from nascent to aging in the marine boundary layer (Carter-Fenk and Allen, 2018). At pH 8.2, the SSA proxy monolayer undergoes a two-dimensional phase transition from a gaseous-tilted condensed (G-TC) coexisting phase to a tilted condensed (TC) phase at 24  $Å^2$ /molecule (lift-off area). After further compression of the TC phase, the monolayer transitioned to the untitled condensed (UC) phase at a surface pressure of 18 mN/m. The maximum surface pressure of the film is around 70 mN/m. At pH 5.6, the lift-off point of monolayer occurs at 25 Å2/molecule, and a surface pressure plateau at collapse occurs at  $\sim 64$  mN/m. The collapse pressure on 0.4 M NaCl at pH 5.6 is slightly lower than that at pH 8.2. When the SSA proxy film is completely protonated on 0.4 M NaCl at pH 2.0, the lift-off area is significantly higher at 28 Å2/molecule, and the  $\pi$ -A isotherm reaches a maximum surface pressure of ~ 58 mN/m before the surface pressure decreases. Combined with Brewster angle microscope (BAM) to visualize the SSA proxy collapse structure, the results showed that at nascent SSA pH, the mixture produced a monolayer with moderate rigidity, which folded when the film was compressed to a collapsed state. However, acidification makes the SSA proxy film to become more rigid and form three-dimensional (3D) nuclei.

In addition, the pH change of the solution also changes the protonation state of the carboxylic acid. Recent work by Carter-Fenk et al. has shown the alginate (a linear anionic polysaccharide) co-adsorption to d31-palmitic acid monolayers, and investigated the effects of different pH values (Carter-Fenk et al., 2021). The carboxylic acid protonation state was varied through the solution pH values of 8.2 and 5.8. Palmitic acid has a reported surface pKa between 8.34 and 8.7, and the pKa values of alginate G and M residues are 3.7 and 3.4, respectively. Thus at seawater pH of 8.2, palmitic acid is partially deprotonated and alginate is fully deprotonated. At pH 5.8, palmitic acid is mostly protonated, and alginate carboxylate groups remain deprotonated. It was found that the protonation state of palmitic acid significantly affected the degree of alginate co-adsorption. For the d31-palmitic acid monolayer at

pH 5.8, alginate co-adsorption degree decreased compared with the chemical system at pH 8.2. Therefore, increased  $d_{31}$ -palmitic acid protonation decreases the extent of alginate co-adsorption.

In our study, we measured the pH of the seawater solution before and after the experiments. The results showed that the natural seawater used had a pH of approximately  $8.13\pm0.02$ , which was measured at about  $8.04\pm0.01$  after exposure to air in the Langmuir trough for about 2 h during the experiment period. That is to say, no matter in the experiment of measuring  $\pi$ -A isotherm or infrared reflection-absorption spectra, the change in pH value of seawater remains at about 0.1, and the overall acidification is relatively slow. Minor changes in pH therefore do not have very strong effects on protonation and deprotonation state of fatty acids. Therefore, we infer that the effect on the  $\pi$ -A isotherm and IRRAS spectral measurements due to changes in seawater pH should not be the dominant factor during the experiment. We have explained the pH changes throughout the experiment in the revised manuscript at page 8:

The pH of natural seawater, initially determined to be about  $8.13\pm0.02$ , was measured to be around  $8.04\pm0.01$  at the end of the experiment.

RC: The partial dissolution of myristic acid most likely accounts for the smaller mean molecular area observed in the proxy mixture isotherm compared to the palmitic acid and stearic acid isotherms (Figure 3). Myristic acid increases the fluidity of the monolayer, thereby expanding the surface pressure-area isotherm when myristic acid remains adsorbed to the surface (see https://doi.org/10.3390/atmos9120503 for further discussion).

AR: In this study, myristic acid (MA) was the shortest of the selected long-chain fatty acids, and the sum of the dispersion interactions produced between the lipids was small. Therefore, the MA film at the air-water interface is more disordered and less tightly packed (Gericke and Huhnerfuss, 1993). The MA is partially soluble in water (Patil et al., 1973), and due to the strong electrostatic interaction between the carboxylate moiety and the water molecules, the desorption rate of MA increases with the deprotonation of the carboxylic acid headgroups. At 20 °C, according to the surface pKa value of 7.88 (Mclean et al., 2005), MA is mostly deprotonated at pH ~8.1, so a stable monolayer cannot be obtained for the natural seawater subphase due to the dissolution phenomenon. In the  $\pi$ -A isotherm experiment, the mechanical forcing from lateral barriers compression also promotes desorption. Therefore, the MA monolayer is unstable and cannot reach collapse. It could also help explain why the  $\pi$ -A isotherm of proxy monolayer mixture occupied a smaller mean molecular area. The partial dissolution of myristic acid may play a role. We have made a further discussion for this phenomenon in the revised manuscript at page 19:

The partial dissolution of myristic acid most likely accounts for the smaller mean molecular area observed in the proxy mixture isotherm compared to the palmitic acid and stearic acid isotherms.

RC: In line 325, the authors mention a "kink point" in the palmitic acid isotherm at  $\sim$ 40 mN/m. Palmitic acid should not have a phase transition at this point. It is possible that this "kink point" is caused by a contaminant that is being squeezed out upon monolayer compression. Does the "kink point" remain upon using a new palmitic acid and chloroform solution? Does the "kink point" disappear when compressing the barriers at 5 mm/min/barrier instead of 3 mm/min?

AR: Before each measurement of the  $\pi$ -A isotherm, we thoroughly cleaned the trough and barriers with reagent alcohol and ultrapure water to ensure contamination-free. Moreover, if the contamination was due to palmitic acid stock solution, the  $\pi$ -A isotherm of the mixed fatty acid would also show a similar turning point near this surface pressure. For mixed fatty acids, however, there was no such twist. Therefore, we can basically rule out the contamination caused by palmitic acid stock solution. Although we cannot clearly explain the reasons for this phenomenon at that time, the  $\pi$ -A isotherm was re-determined using freshly prepared palmitic acid solution and the results are shown in Figure 3 in the revised manuscript. For the  $\pi$ -A isotherm of the newly determined palmitic acid, we found that its  $\pi$ -A isotherm with stearic acid and mixed fatty acids does exhibit similar collapse behavior at a surface pressure of about 50 mN/m. We have revisited this section in the revised manuscript at page 18-19:

After experiencing a kink point at  $\sim$ 48 mN m-1, it continues to rise to the maximum surface pressure of  $\sim$ 57 mN m-1 and collapses. Both the lift-off area and molecular area of the stearic acid film decreased more than those of palmitic acid film. This is caused by the fact that the interaction (van der Waals force) between the molecules increases as the molecular weight of long chain fatty acid increases. That is, increased attraction leads to a decrease in distance between SA molecules.

The partial dissolution of myristic acid most likely accounts for the smaller mean molecular area observed in the proxy mixture isotherm compared to the palmitic acid and stearic acid isotherms. Moreover, we found that the  $\pi$ -A isotherms of mixed fatty acids exhibit similar collapsing behavior to those of stearic acid and palmitic acid at a surface pressure of about 50 mN m-1.

Figure 3.  $\pi$ -A isotherms of myristic acid, palmitic acid, stearic acid and mixed fatty acids. The black trace represents the background natural seawater solution with no fatty acid spread.

Regarding the barrier compressing speed, the 3 mm/min we chose for this study was determined based on our previous pre-experimental results. We have used stearic acid film as an example to investigate the effect of barrier moving speed on its  $\pi$ -A isotherms. When the amount of organic solution was 50 µL and the volatilization time was 15 min, the two barriers compressed the stearic acid molecules at a uniform speed of 1, 3, and 6 mm/min to obtain  $\pi$ -A isotherms of their monolayers. It was found that the isotherms obtained at 1 and 3 mm/min were very close, and when the barrier compressing speed reached 6 mm/min, the entire isotherm shifted to the right. When the barrier compressing rate is slow enough, it can be assumed that the film is in equilibrium state, while the surface pressure value obtained in the non-equilibrium state is obtained when compressed quickly. Therefore, a barrier moving speed of 3 mm/min is appropriate for this study.

Figure R1. The influence of barrier moving speed on  $\pi$ -A isotherms of stearic acid monolayer.

RC: In line 206, the barrier compression speed should be specified as 3 mm/min/barrier (if that is the case).

AR: Thanks for the Referee's reminder. We have modified it in the revised manuscript at page 11:

The barriers were compressed at a rate of 3 mm min-1 per barrier and isotherm data were collected for surface pressure  $\pi$  (mN m-1) versus area per molecule (Å2).

RC: In lines 353-355, the authors cite Vazquez de Vasquez et al., 2022 (https://doi.org/10.1021/acsearthspacechem.2c00066) for saccharide concentrations in the ocean. The authors should instead cite the original papers for these measurements: https://doi.org/10.1016/0304-4203(92)90020-B, https://doi.org/10.1021/cr500713g, and https://doi.org/10.1021/acsearthspacechem.9b00197. However, Vazquez de Vasquez et al. corroborate the authors' argument that the saccharides interact with the monolayer headgroups and expand the monolayer. Additionally, Vazquez de Vasquez et al. argue that glucuronate intercalates into a stearic acid monolayer. Thus, a brief discussion and/or statement on the Vazquez de Vasquez et al. results is warranted in the context of this manuscript's conclusions on the saccharide-carboxylic acid hydrogen bonding interactions. This statement/discussion would perhaps fit in with the discussion in lines 368-375.

AR: We have cited the original literature recommended by the Referee in the revised manuscript at page 20:

Considering that saccharides in the ocean represent approximately 20% of the dissolved organic carbon (Pakulski and Benner, 1992; Hasenecz et al., 2019), the saccharide concentration is about  $0.14-0.20 \text{ mg L}^{-1}$ .

Vazquez de Vasquez et al. investigated the co-absorption of alginate and its representative monomeric form glucuronate to a stearic acid monolayer as a function of saccharide concentration on an ocean proxy solution (De Vasquez et al., 2022). Their experimental results showed that for glucuronate, the film reached a maximum surface pressure in the range  $\sim 60-65$  mN/m, while the mixed proxy film reached a maximum surface pressure between 50 mN/m and 60 mN/m in our experiments. In the case of glucuronate, they also observed that the monolayer expansion is not monotonic. The drastic expansion, or an increase in the surface area of individual molecules increases disproportionately, which strongly indicates that it exists at the interface where it destroys the packing of the monolayer. However, no such disproportionality was observed for the two saccharides we studied. As the concentration of glucuronate increases, small changes in the brightness of the BAM image also point to the glucuronate intercalation into the monolayer, producing reorganization. Using Langmuir isotherms, surface-sensitive infrared reflection-absorption spectroscopy and Brewster angle microscopy, they demonstrate that glucuronate may intercalate and induce significant reorganization within the monolayer. Data on glucuronate co-adsorption to the deprotonated monolayer suggest that glucuronate surface propensity is a contributor to saccharide composition. The results of Vazquez de Vasquez et al. have been further discussed in a comparative way in the revised manuscript at page 21-22:

De Vasquez et al. (2022) also demonstrated that glucuronate interacts with and expands the stearic acid monolayer. Furthermore, they suggested that glucuronate intercalates into the stearic acid monolayer and leads to monolayer reorganization. Spectral evidence is needed to further clarify whether intercalation occurs in our study.

RC: Lines 380-383: Due to the partial solubility of myristic acid at seawater pH, it is possible that higher concentrations of glucose or trehalose simply decrease the myristic acid solubility due to competitive hydration. In other words, the saccharides are weakly "salting out" the myristic acid from the seawater, enhancing myristic acid adsorption at the air/water interface and expanding the monolayer. I recommend conducting a control experiment in which the surface pressure of myristic acid alone is monitored as a function of saccharide concentration. Spread the same amount of myristic acid on the seawater surface, and test whether higher concentrations of

saccharides increase the surface pressure (increase myristic acid adsorption). Normalize the change in surface pressure to any changes in the subphase surface tension due to the different concentrations of saccharides in the seawater. Alternatively, the authors could use deuterated myristic acid and track the C-D vibrational modes with IRRAS as a function of saccharide concentration. If the overall intensity of the C-D modes do not change with increasing saccharide concentration, then the saccharides are not impacting myristic acid adsorption at the air/water interface.

AR: As the Referee noted, high concentrations of glucose or trehalose do have the potential to "salt out" myristic acid weakly from seawater and reduce its solubility. Therefore, the adsorption of myristic acid at the air/water interface was enhanced and the monolayer was expanded. According to the Referee's suggestion, we supplemented a set of controlled experiments in which the surface pressure of myristic acid alone is monitored as a function of the same saccharide concentration. As shown in Figure R2, we found that high concentrations of glucose do not increase the surface pressure, but all manifest as a decrease in surface pressure. In particular, the decrease in surface pressure is most significant when the maximum concentration of glucose is 5 g L-1. Therefore, enhanced surface adsorption of myristic acid due to salting out may not have played an important role in this study.

---

## Author Comment (AC3)

**Authors' Response to Reviews of**

"Insoluble lipid film mediates the transfer of soluble saccharides from the sea to the atmosphere: the role of hydrogen bonding"

Minglan Xu, Narcisse Tsona Tchinda, Jianlong Li, Lin Du*

We thank the Referee for the constructive comments. We have addressed the comments point by point below and modified the manuscript accordingly. For clarity, the Referee's comments (RC) are reproduced in blue, authors' responses (AR) are in black and changes in the manuscript are in red color text.

**Anonymous Referee #3**

RC: The manuscript under consideration presents a compelling model for describing the transfer of saccharides into the aerosol phase through artificial breaking waves. The authors use a combination of state-of-the-art methods to support their model: SMPS, Langmuir isotherms, PM-IRRAS, and TEM imaging all serve as a basis for exploring the interactions within a ternary system of saccharides, seawater and insoluble fatty acids. The scientific arguments are logically sequenced, and well-referenced to previous work in the discipline. Overall, I expect that this work will be well-received by the scientific community as it provides interesting insight into the transfer of organic material from the SML into the aerosol phase.

My main criticism of the work in its present form is the authors' use of PM-IRRAS to elucidate the role of hydrogen bonding between the saccharide and fatty acid layer. In particular, I would like to see stronger evidence that there was a shift in the $\nu(C=O)$ frequency, as it is not abundantly clear from Figure 5 in its present form. In addition, there are some finer points of the authors' scientific arguments that could be expanded upon. I think that these should be addressed before the manuscript is accepted, as it helps to contextualize their results.

AR: We are sorry that the original IRRAS spectra are not particularly clear. To this end, we specially performed Gaussian fitting of the collected IRRAS spectra in the revised manuscript, so as to achieve a better resolution of each peak. In addition, we summarize the properties of each peak, such as wavenumbers, reflectance-absorbance intensities, peak areas and full width at half maximum (FWHM, cm$^{-1}$) in a separate table and provide it in the supplement. Therefore, we can now clearly distinguish and identify variations such as the shift and intensity of each peak to better support our results analysis. In view of the results and discussion, we have reconsidered, and carried out more detailed analysis and discussion in some sections.

Additionally, we also showed the wavenumbers, reflectance-absorbance intensities,

peak areas and full width at half maximum (FWHM, cm⁻¹) values of each fitted peak in Table S2 in the supplement. In this work, the relatively low frequencies of $\nu_{as}(CH_2)$ (2916–2918 cm⁻¹) and $\nu_s(CH_2)$ (2848–2851 cm⁻¹) hint that the molecular conformation of the fatty acid alkyl chains is dominated by the highly ordered *all-trans* conformation (Li et al., 2019).

[Figure]

Figure 5. PM-IRRAS spectra (3000–2800 cm⁻¹) of mixed fatty acids at the air/seawater interface at different (a) glucose, and (b) trehalose concentrations in the subphase.

Carboxylic acids possess one hydrogen bond donor (hydroxyl) and one hydrogen bond acceptor (carbonyl) within the same functional group, the carboxyl group. The carbonyl stretching modes ($\nu(C=O)$) of the carboxyl group at ~1734 cm⁻¹ (unhydrogen bonded), 1725 cm⁻¹ (singly hydrogen bonded) and 1708 cm⁻¹ (doubly hydrogen bonded) were observed in seawater (Gericke and Huhnerfuss, 1993), with the strength at 1734 cm⁻¹ being the highest (Figure 6). This band component at 1734 cm⁻¹ is put down to the conformation with the carbonyl group almost parallel to the water surface and the hydroxyl group is oriented toward the water surface, which is not conducive to the formation of hydrogen bond with water subphase (Muro et al., 2010). For saccharide concentrations ranging from 0.1 to 2 g L⁻¹, the unhydrated C=O band was observed to be depressed, and the singly and doubly hydrogen bonded carbonyl components at ~1720 and ~1708 cm⁻¹ became dominant (Johann et al., 2001). At the highest glucose concentration, the Langmuir model appears to capture a saturation effect, where the establishment of hydrogen bonds is associated with a strong initial increase in glucose organic enrichment, followed by surface saturation at higher organic concentration. We also displayed the wavenumbers, reflectance-absorbance intensities, peak areas and full width at half maximum (FWHM, cm⁻¹) values of each fitted peak in the region of 1800–1300 cm⁻¹ in Table S3 in the supplement.

The broad and strong antisymmetric carboxylate stretch ($\nu_{as}(COO)$) were observed at ~1564 cm⁻¹, and the symmetric carboxylate stretch ($\nu_s(COO)$) at ~1415 cm⁻¹. The presence of salt in seawater caused the $\nu_{as}(COO)$ to split into three peaks at ~1564, ~1544 and ~1528 cm⁻¹. Additionally, we found a shift in the major carboxylate

stretching mode from 1564 to higher frequency ~1572 cm$^{-1}$, which may be indicative of carboxylate dehydration upon interactions with saccharides. Another distinctive feature in all spectra obtained at ~1469 cm$^{-1}$ was assigned to the CH$_2$ scissoring vibration ($\delta$(CH$_2$)) of the aliphatic chain (Muro et al., 2010).

[Figure]

Figure 6. PM-IRRAS spectra (1800–1300 cm$^{-1}$) of mixed fatty acids at the air/seawater interface at different (a) glucose, and (b) trehalose concentrations in the subphase.

RC: **Line 35:** "SSA represents **the** major source of aerosol particle populations". I think this is a complicated assertion to make. While SSA emission per annum is the greatest of all sources with respect to mass (Textor et al, 2006), the same can't be said about number: even in the Southern Ocean, where sea spray production is rampant, SSA is outnumbered by sulfate aerosols (Quinn et al, 2017). As the sentence goes on to describe effects relating to CCN and IN, I think the statement should be softened to "SSA represents **a** major source of aerosol particle populations".

Textor, C., Schulz, M., Guibert, S., Kinne, S., Balkanski, Y., Bauer, S., Berntsen, T., Berglen, T., Boucher, O., Chin, M. and Dentener, F., 2006. Analysis and quantification of the diversities of aerosol life cycles within AeroCom. Atmospheric Chemistry and Physics, 6(7), pp.1777-1813.

Quinn, P.K., Coffman, D.J., Johnson, J.E., Upchurch, L.M. and Bates, T.S., 2017. Small fraction of marine cloud condensation nuclei made up of sea spray aerosol. Nature Geoscience, 10(9), pp.674-679.

AR: Indeed, in many studies, SSA is considered to be one of the largest sources of primary aerosol particles in the atmosphere on a mass concentration basis. For example, Textor et al. analyzed the sources of sea salt (SS), dust (DU), sulfate (SO4), black carbon (BC) and particulate organic matter (POM) as simulated by sixteen global aerosol models in the framework of the AeroCom intercomparison exercise (Textor et al., 2006). The total all-models-average aerosol source amounts to 18800 Tg/a. Sources are dominated by SS with 16600 Tg/a, followed by DU (1840 Tg/a), SO4 (179 Tg/a), POM (96.6 Tg/a), and finally BC (11.9 Tg/a) (Figure R1).

[Figure]

Figure R1. Global, annual average emissions [Tg/a] in all models for DU, SS, SO4, BC, POM, and AER (=total dry aerosol).

Due to the large proportion of SSA in the global atmosphere, especially considering marine haze and cloud layers, it can have a great impact on cloud formation and atmospheric radiation balance. Although SSA has very high cloud condensation nuclei (CCN) activation potential, the majority of its population, residing in submicron sizes, tend to be obscured by non-sea spray CCN. Recent approaches to estimate the submicron SSA employed a free-monomodal lognormal analysis that depicts the global oceanic CCN population comprising less than 30% SSA. Xu et al. (2022) derived the SSA distribution from a unique five-year data set on aerosol microphysics and hygroscopicity in the Atlantic Ocean. The method takes advantage of the unique ultra-high hygroscopicity of inorganic sea salt and can identify the submicron sea spray down to 35 nm with high time and size resolution. In stark contrast to previous studies, the hygroscopicity coupled multimode fitting analysis yields SSA-derived CCN that is 500% higher than the estimated results obtained using the free monomodal method. That said, the contribution of SSA to global CCN, particularly the Aitken mode SSA, may be overlooked.

Hence, the global atmospheric impact of SSA is quite complex. We have softened the statement to "SSA represents **a** major source of aerosol particle populations" in the revised manuscript at page 3:

Sea spray aerosol (SSA) represents a major source of aerosol particle populations and significantly impacts the earth's radiation budget, cloud formation and microphysics by serving as cloud condensation nuclei (CCN) and ice nuclei (IN), and microbial cycling (Bertram et al., 2018; Partanen et al., 2014)

RC: **Line 152**: "surface seawater was obtained by dipping an HDPE container through the seawater surface." I think it would be misleading to describe your collection method as sampling only surface water as this manuscript often references the SML, which is <1 mm thick. There are specific glass plate sampling methods for collecting SML which would have required a glass plate. The method you described (which is fine, in principle), might be better described as having collected both "surface and near-surface seawater."

AR: Different types of seawater samples (such as sea-surface microlayer, subsurface seawater, surface water and bulk water) have different sampling methods. There are three main ways to collect surface seawater: sampler sampling; pumping water samples; by means of adsorption, ion exchange or electrodeposition, the elements or compounds to be measured are enriched and sampled on site. Among them, the sampler sampling method is more common. The general requirements of the sampler can make the water inside and outside the bottle quickly and fully exchange; the closing system is sealed reliably; the material has corrosion resistance, non-wetting sewage sample and non-adsorption of components to be tested; sampler should not be too heavy. Before using polyethylene barrel for sampling, rinse the barrel with water samples 2~3 times. During sampling, the mouth of the barrel is immersed into the water facing the direction of the water flow. After the barrel is filled with seawater, it is quickly raised to the surface to avoid floating substances on the surface from entering the sampling barrel. The surface seawater collected in our study refers to the seawater collected within 0.1~1 m below the sea surface.

For SML samples, the screen sampling method, glass plate sampling method and rotating cylinder sampling method are often used. Among them, the glass plate sampling method is the most widely used. The glass plate method is to immerse a certain specification of flat glass vertically into the water surface, and then lift it vertically from the water at a certain speed, a certain thickness of SML remains on the glass plate, with the scraper to scrape the residue into the sampling bottle. The general sampling thickness of the glass plate sampler is 40-100 μm, the sampling thickness of the glass plate sampler is ideal and can meet the conditions.

We have provided a more comprehensive description of the collected surface seawater in the revised manuscript at page 8:

Here, surface seawater (within 0.1~1 m below the sea surface) was obtained from a pier on the coast by immersing high-density polyethylene containers into the water. The sampled seawater was microfiltered through 0.2 μm polyethersulfone filter (Supor®-200, Pall Life Sciences, USA) to remove large particles such as sediments, algae and bacteria. The filtered seawater was used for SSA generation and as a filling subphase for interfacial experiments. The pH of natural seawater, initially determined to be about 8.13±0.02, was measured to be around 8.04±0.01 at the end of the experiment.

RC: **Line 252:** I think you need to provide more evidence that plunging jets are similar to breaking waves. The previous work that you cited (Christiansen et al, 2019) does not provide any data on the bubble size distribution, nor does the reference work for your apparatus (Liu et al, 2022). In particular, Prather et al (2013) only described similarities between real breaking waves and plunging sheets (which are different from plunging jets). Looking at a similar plunging jet apparatus described by Salter et al (2014) shows that the bubble size distribution produced by a plunging jet is broadly similar to the plunging sheet shown in Prather et al (2013) and Stokes et al (2013). However, note that the exponents of the power law described by Stokes et al (2013) for the plunging sheet apparatus and real waves are larger than for the plunging jet described by Salter et al (2014) (See Table 4 in Salter and Figure 4 in Stokes). Thus, I would suspect that your bubble size distribution was much broader than for true breaking waves. This ought to be discussed with slightly more nuance in the present manuscript. Larger bubbles have a smaller surface-area-to-volume ratio, which ultimately influences the relative production of film drops versus jet drops. Jet drops, whose composition is more strongly tied to the subsurface below the SML are likely depleted in OM.

Salter, M.E., Nilsson, E.D., Butcher, A. and Bilde, M., 2014. On the seawater temperature dependence of the sea spray aerosol generated by a continuous plunging jet. Journal of Geophysical Research: Atmospheres, 119(14), pp.9052-9072.

Stokes, M.D., Deane, G.B., Prather, K., Bertram, T.H., Ruppel, M.J., Ryder, O.S., Brady, J.M. and Zhao, D., 2013. A Marine Aerosol Reference Tank system as a breaking wave analogue for the production of foam and sea-spray aerosols. Atmospheric Measurement Techniques, 6(4), pp.1085-1094.

AR: The main mechanism of SSA production is bubble-mediated, when bubbles generated by breaking waves burst at the surface. The bursting process produces two types of droplets: film droplets and jet droplets. Film drops are formed when the film of the bubble cap bursts, and jet drops form when the vertical water capillary collapses due to gravity. It is known that the size of the parent bubble determines the number of film droplets and jet drops produced: the large bubble mainly produces film drops, while the small bubble mostly produces jet drops (Woolf et al., 1987). Film drops are responsible for the major proportion (~60%–80%) of submicron particles, whereas jet drops mostly contribute to the production of supermicron particles (Wang et al., 2017). A recent study by Jiang et al. (2022) reported observations of a flapping shear instability mechanism that results in a significant fraction of submicron aerosols produced by jet drops from very small (~ 1 mm radius) bubbles.

When producing SSA in laboratory conditions, the challenge is to simulate the characteristics of key processes for bubble-mediated aerosol production in the real environment. Commonly applied methods include atomizers and bubbling tanks with

sintered glass diffusers or water jet bubbling systems, the plunging sheet in the marine aerosol reference tank (MART) system and laboratory breaking wave (Christiansen et al., 2019; Salter et al., 2014; Fuentes et al., 2010; Stokes et al., 2016; Prather et al., 2013). Aerosol atomizers, being widely used in the laboratory to produce aerosol mixtures, cannot mimic the dynamics of the marine bubble bursting, whereas this process can be better replicated in a bubbling tank.

Stokes et al. (2013) compared the size distribution of bubbles in the MART plumes with bubbles produced by sintered glass filters and oceanic and laboratory wave channel distributions. The glass filter was set at a depth of about 25 cm (filter surface to water surface) and the dried nitrogen gas ($0.5 \text{ L min}^{-1}$) was forced through four filters, two 90 mm diameter E-filters and two 25 mm diameter A-filters. The plunging sheet peak flow rate was about $1 \text{ L min}^{-1}$, falling through a height of approximately 10 cm, and modulated with on/off times of about 4 s on and 10 s off. The breaking wave was operated within a sealed 33 m wave channel filled with natural seawater pumped directly from the Pacific Ocean, with continuous wave breaking at 0.6 Hz. As can be seen from Figure R2, similar bubble spectra are produced using plunging sheet and breaking waves, while different bubble size spectra are produced by sintered glass filters. Importantly, the measured bubble spectrum of the breaking waves matches the shape and Hinze scale of the bubble spectrum of the previously measured open ocean breaking waves (Deane and Stokes, 2002). Previous studies using plunging jets have produced a similar bubble size distribution only up to a radius of 0.57 mm (Fuentes et al., 2010).

[Figure]

Figure R2. Intercomparison of bubble size distributions from a laboratory breaking wave, the plunging sheet in the MART system and two distributions from sintered glass filters.

King et al. (2012) measured the size spectra of bubbles produced by diffuser and plunging jet using a mini-BMS (bubble measurement system). The size spectrum of bubbles produced by a diffuser at an air flow of 1.5 L min⁻¹ and at a depth of 26.5 cm below the water surface shown in Figure R3 represents the production of this type of bubble at an air flow of 0.25 to 2 L min⁻¹. Using two nozzle sizes (4 mm and 16 mm) to form the plunging jet, the resulting bubble size spectra are shown in Figure R2, along with the power law exponent corresponding to the descending portion of each spectrum. Similar to the results of previous studies on sea spray tank studies (Fuentes et al., 2010; Sellegri et al., 2006; Hultin et al., 2010), they find that the shape of the bubble spectra produced by the jet is more similar to the oceanic bubble size spectra, as determined by the power law exponent. By visualizing the bubble size distribution of plunging jets and diffusers with that of ocean bubbles, they conclude that plunging jets better simulate plunging breaking waves in terms of bubble plume characteristics.

[Figure]

Figure R3. Bubble size distributions produced using the diffuser and the plunging jet in artificial seawater having a salinity (TMSS) of 35‰.

Arguably, the closest reproduction of wave breaking and bubble generation processes was achieved in the ocean–atmosphere facility (33 m wave channel) of Prather et al. (2013); however, the complexity and costs of the experiments were rather high. Additionally, high-speed photography has enabled a detailed description of the bubble rupture process. Unfortunately, we were not able to photograph the bubbles with a high-definition camera and calculate their sizes in our study. While the aerosol generation technique used in this study cannot come close to fully simulating real SSAs, it provides a controlled framework for examining specific chemical and physical properties that contribute to saccharide transfer in marine systems.

In the revised manuscript at page 13-14, we have made necessary supplementary explanations on the plunging jet method used.

The submicron particle size distributions produced by the plunging jet generator are well represented by lognormal mode. In the absence of saccharide, a broad, unimodal mode of the particle size distribution around 168 nm was generated. This observation agrees quite well with a previous study that produced SSA by the plunging jet method with the mode of the particle size distribution ~162 nm (Christiansen et al., 2019). Moreover, the SSA yielded by plunging waterfall also has a size distribution similar to that yielded by the breaking wave, which particle number size distribution is ~162 nm (Prather et al., 2013). This contrasts with most previous laboratory studies using sintered glass filters or frits, which tend to exhibit a smaller mean diameter and narrower distribution. This may be expected, given that similar bubble size distributions exist in the two generation mechanisms using plunging waterfall and breaking waves. A previous study using plunging jets has produced similar bubble size distributions (Fuentes et al., 2010). Importantly, the measured bubble spectrum for the breaking waves matches the shape and Hinze scale of the bubble spectra of the previously measured open ocean breaking waves (Deane and Stokes, 2002). Although we did not directly measure the bubble spectra generated by the plunging jet method in this study, it should be able to better simulate the properties of breaking waves according to the above empirical studies. Moreover, we compared the particle size distributions of SSA generated in our laboratory with those measured in field studies (Quinn et al., 2017; Xu et al., 2022). As shown in Figure S3, it was observed that the size distribution of both laboratory-generated SSAs and SSAs measured in the field had a major accumulation mode in the range of ~111–172 nm. However, the number concentration of SSAs produced in our experiment is about 2 orders of magnitude higher than that in the real environment. As a result, the jet sea spray generator system is capable of a wide range of measurements (e.g., size-resolved hygroscopicity and heterogeneous reactivity) that are not achievable at low number concentrations.

RC: **Line 287:** You describe the stability of the surface layer in the presence of fatty acids, but you are constantly disrupting the surface with your plunging jet which is mixing the SML into the subsurface waters. You describe later on (Line 312) that the collapse of the 2D film is itself an irreversible process. Part of my concern with your sampling method is that it does not allow for any transient redevelopment of the SML. There is a time constant related to the development of the SML after being perturbed. In the real ocean, waves rarely ever break the same surface twice. Plunging sheet methods (Stokes et al, 2013) and wave chambers (Prather et al, 2013) allow for the redevelopment of an SML between wave-breaking events. I think it is worth discussing within your manuscript that the transfer of saccharides to the aerosol phase may actually have been limited by the continuous mixing of the SML into the subsurface.

AR: Natural SSAs are mainly produced by whitecaps in the ocean, which are episodic in nature. The visible white area on the sea surface during and subsequent to a

wave-breaking episode is due to the presence of foam, a collection of bubbles floating at the air-sea interface, each separated by a thin liquid film. The persistence of whitecap foam, as measured by its exponential decay time, is mainly in the range of 2-4 s, with occasional extensions up to 10 s (Callaghan et al., 2012). Collins et al. (2014) conducted foam production experiments using the plunging-waterfall mechanism in a MART system to investigate the effects of pulsed versus continuous foam production. The plunging waterfall was operated in both "continuous" and "pulsed" modes. In "continuous" mode, water was circulated through a centrifugal pump from the bottom of the tank to a waterfall device suspended above the water surface, creating a continuous waterfall. In "pulsed" mode, the recirculation flow to the waterfall apparatus was modulated with a 4 s on, 4 s off pattern. During the "on" cycle, the flow rate of water was approximately 40 L/min. The effect of continuous bubble production on the composition of SSA, which can lead to the accumulation of foam at the water surface several bubble layers thick, is directly discussed.

The number size distributions of SSA particles were observed to be nearly identical between the pulsed and continuous plunging protocols when implemented using unamended natural seawater (Figure R4(a)). The change in the shape of the size distribution is clearly evident between the continuous and pulsed plunging cases when the seawater is enriched with organic matter. The concentrations of particles with $d$p > 0.3 and $d$p < 0.05 μm are smaller while concentrations of particles with $d$p = 0.05–0.125 μm were higher during continuous plunging (Figure R4(b)). Continuous plunging resulted in a tank-wide layer of foam that accumulated on the water surface, whereas surface foam had a patchy character when the plunging waterfall was pulsed at 4 s intervals. The particle concentration with $d$p > 0.3μm was decreased during continuous plunging, which may be due to the weakening of jet droplet production. The presence of the foam layer on the seawater surface may be capable of prohibiting or curtailing jet droplet production by assimilating rising bubbles into the foam layer before bursting. The presence of a significant surface foam layer seemed to enhance the production of SSA with $d$p =0.05–0.125 μm, suggesting that cap film rupture plays an important role in the production of SSA particles within this limited size range. However, the organic carbon measured in SSA produced by continuous plunging was substantially higher than pulsed plunging in organic enriched seawater. It can also be called over-expression of organic matter in the SML. The differences observed in SSA composition between the pulsed and continuous plunging modes underscores the importance of preserving the transient nature of surface foam inherent to the wave breaking process in the production of SSA in the laboratory when concentrations of organic matter in the seawater are elevated.

Wurl et al. (2011) showed that the SML exists on the ocean surface for wind speeds up to 10 m s$^{-1}$ (global ocean mean wind speed is approximately 6 m s$^{-1}$), so the existence of the SML is relevant for many instances of wave-induced bubble and foam production. At the same time, dynamic physical processes at the ocean surface can exert control on the thickness and extent of the SML (Cunliffe et al., 2013). In the

plunging waterfall mechanism, the mixing of seawater surface material back into the water column is a phenomenon that counteracts bulk-to-surface transport of surface-active organic matter by the rising bubble plume. Therefore, organic enrichment in SML can be similarly mitigated using a technique of mixing surface-active organic materials into the return water column to generate aerosols. This is in good agreement with the plunging jet technology used in our experiment to generate SSA. Unfortunately, the numerical quantification of saccharide transfer from seawater to SSA was not achieved in our study. Therefore, we did not discuss much about mixing the SML into the water column. I believe that in our future research, special attention will be paid to this.

[Figure]

Figure R4. Number size distributions for MART-generated SSA particles using continuous (red) and pulsed (black) plunging-waterfall modes (±1σ error bars).

In view of the plunging jet technology, we have made a further discussion in the revised manuscript at page 16-17:

When the fatty acid surfactant was added to seawater alone, the number concentration decreased by about 17.2%, while the presence of glucose resulted in a decrease of about 21.6%. Moreover, fatty acids showed the highest inhibitory effect on SSA produced by trehalose-containing seawater solution, whose concentration decreased by about 49.4%. We ascribe that the surface layer is significantly more stable in the presence of fatty acids, even when disturbed by the plunging jet, thus resulting in less bubble bursting. Furthermore, the continuous plunging caused a layer of foam to accumulate on the surface of the water. The presence of the foam layer on the seawater surface may be capable of prohibiting the production of droplets by assimilating rising bubbles into the foam layer before bursting.

RC: **Section 3-2:** I just wanted to comment that I found this entire section well-written and illuminating.

AR: We thank the Referee for such a positive comment.

RC: **Lines 462-465:** Here you are describing a shift in the vibrational frequency as evidence of hydrogen bonding. While this is not my specific area of expertise, I am having a hard time seeing a systematic shift in the peak of v(C=O) in either Figure 5a or b. Unless I am gravely misinterpreting these plots, the peak appears to go back and forth between the dashed lines you highlighted as the saccharide concentration increased, rather than one peak systematically outweighing the others as the concentration increased. Case in point, the dominant peak for the carbonyl stretch mode v(C=O) appears to be 1732 cm$^{-1}$ for both seawater AND your highest concentration of Glucose in Figure 5a. Perhaps you could add an inset to Figure 5 that zooms in on this band and better describes the phenomena you are observing. This is a key observation that you repeatedly use throughout the remainder of the manuscript to support evidence of hydrogen bonding between the saccharide and fatty acid. It ought to be crystal clear to the reader.

AR: We are sorry that the original IRRAS spectra do not provide a good way to distinguish changes in peak position in the carbonyl region. In order to make better identification, we carried out Gaussian fitting based on the measured IRRAS spectra, and redrew the infrared spectra to display in the revised manuscript. In addition, we also added a table in the supplement to summarize the wavenumbers, reflectance-absorbance intensities, peak areas and full width at half maximum (FWHM, cm$^{-1}$) values of each fitted peak. Through this information, we can better compare the changes of carbonyl mode under different saccharide concentrations. In the revised manuscript, we have discussed the changes of carbonyl in different situations in more detail.

The carbonyl stretching modes ($v$(C=O)) of the carboxyl group at ~1734 cm$^{-1}$ (unhydrogen bonded), 1725 cm$^{-1}$ (singly hydrogen bonded) and 1708 cm$^{-1}$ (doubly hydrogen bonded) were observed in seawater (Gericke and Huhnerfuss, 1993), with

the strength at 1734 cm⁻¹ being the highest (Figure 6). This band component at 1734 cm⁻¹ is put down to the conformation with the carbonyl group almost parallel to the water surface and the hydroxyl group is oriented toward the water surface, which is not conducive to the formation of hydrogen bond with water subphase (Muro et al., 2010). For saccharide concentrations ranging from 0.1 to 2 g L⁻¹, the unhydrated C=O band was observed to be depressed, and the singly and doubly hydrogen bonded carbonyl components at ~1720 and ~1708 cm⁻¹ became dominant (Johann et al., 2001). At the highest glucose concentration, the Langmuir model appears to capture a saturation effect, where the establishment of hydrogen bonds is associated with a strong initial increase in glucose organic enrichment, followed by surface saturation at higher organic concentration. We also displayed the wavenumbers, reflectance-absorbance intensities, peak areas and full width at half maximum (FWHM, cm⁻¹) values of each fitted peak in the region of 1800–1300 cm⁻¹ in Table S3 in the supplement.

[Figure]

Figure 6. PM-IRRAS spectra (1800–1300 cm⁻¹) of mixed fatty acids at the air/seawater interface at different (a) glucose, and (b) trehalose concentrations in the subphase.

RC: **Lines 487-488:** Again, I had to look quite closely to see the trifurcation of the vas(COO) peak. This is more obvious upon closer inspection than my previous comment about v(C=O), but an inset of Figure 5 that focuses on the 1500-1600 cm⁻¹ region might be helpful to the reader.

AR: Like in our previous reply, we also performed Gaussian fitting on the peaks in the carboxylate region. The new IRRAS spectra presented in the revised version can better resolve the carboxylates peaks. The wavenumber changes of carboxylate modes at different saccharide concentrations are also discussed with more nuance in the revised manuscript at page 28:

The broad and strong antisymmetric carboxylate stretch ($v_{as}$(COO)) were observed at ~1564 cm⁻¹, and the symmetric carboxylate stretch ($v_s$(COO)) at ~1415 cm⁻¹. The

presence of salt in seawater caused the $v_{as}$(COO) to split into three peaks at ~1564, ~1544 and ~1528 cm$^{-1}$. Additionally, we found a shift in the major carboxylate stretching mode from 1564 to higher frequency ~1572 cm$^{-1}$, which may be indicative of carboxylate dehydration upon interactions with saccharides.

RC: **Figure 6:** This is a beautiful figure, but one of my concerns is that you have analyzed (and are thus comparing) particles of different sizes. There are many studies which suggest that the fraction of organic matter within the generated aerosol can be highly size-dependent for particles produced from the same bulk water composition. This complicates your comparison somewhat and ought to be discussed with more nuance in this section; particularly, as you reference Estillore et al (2017)'s finding that the core-shell morphology is highly dependent on the salt-organic ratio. I think that your qualitative argument is fine, but some additional citations and discussion of the inherent limitations of comparing different-sized particles are needed.

AR: Our original intention in acquiring TEM images was actually to examine the particle morphology and qualitatively compare the SSA differences between different model systems. TEM images were used as auxiliary analytical means to show the exact transfer of saccharides from bulk seawater to SSA particles, and as organic components to form the core portion of the core-shell morphology SSA. The analysis and comparison of the ratio of shell layer and inorganic salt core among different model systems by TEM images are beyond the scope of our current work. We have reorganized the discussion on TEM images in this section.

However, as shown in Figures 7d–f, the presence of fatty acid layer on the surface not only reduces the number concentration of SSA produced but also tends to maintain the cubic shape of the core of SSA. When fatty acids and saccharides coexist, we can still observe the preservation of core-shell structure.

[Figure]

Figure 7. TEM images of morphology identified for sea spray aerosols produced from (a) natural seawater, (b) seawater with glucose and (c) seawater with trehalose

without fatty acids organic layer; (d) natural seawater with fatty acids, (e) seawater with glucose and fatty acids, (f) seawater with trehalose and fatty acids.

RC: **Line 575:** "poor". I think this is a bit of a harsh way of phrasing the scope of this study. Suggest softening "poor" to "limited".

AR: We have made changes in the revised manuscript at page 31:

Other limitations to this study include the limited representation, by the simple chemical structural models, of the myriad complex biomolecules that exist in the ocean, spanning dissolved, colloidal and particulate matter.

RC: **Line 582-585:** While this is the general view, there is some nuance to this assertion specifically for CCN. The hygroscopicity of a composite aerosol is generally well-modelled according to a linear mixture model based on volume fraction (Petters and Kreidenweiss, 2007). The hygroscopicity of your aerosol was likely between that of glucose (k=0.17; Ziemann, Kreidenweiss and Petters, 2013) and that of sea salt (k=1-1.25; Zieger et al, 2017). Further, you combine De Vasquez et al (2022) and (Quinn et al 2015; Hasenacz et al 2019) to conclude that the oceanic concentration of saccharides is just 0.14 mg/L, which is substantially lower than the concentrations observed here. So, how considerable of an effect is this going to have on hygroscopicity? Here is some back-of-the-envelope math:

Ocean Salinity (g/L): 35

Bulk Saccharide Concentration (g/L): 0.00014

Density of Glucose (g/cm3) ~ 1.56

Density of Salt (g/cm3) ~ 2

Enrichment factor (Zeppenfeld et al, 2021): <167000

Mass Ratio of saccharide in aerosol (g/g): (0.00014/35)*167000 = 0.67

Volume Ratio (L/L): 0.67*2/1.5 = 0.89

k = 0.89*0.17 + 0.11*1.1 = 0.27

This is likely a lower limit of the resulting hygroscopicity of your mixed aerosol since it assumed that the enrichment factor is on the largest end of the factors reported by Zeppenfeld et al (2021). Relating this to the sc-Dd curve presented by Petters and Kreidenweiss (2007) in Figure 2, a supersaturation of just 0.1% is required to activate >50% of your particle size distribution as CCN. Consider that 0.1% is the lower end of supersaturations experienced during cloud formation and consider that the

calculation above is likely an upper limit of the abundance of the saccharide within the aerosol. Case in point, at a supersaturation of 1.0% virtually your entire particle size distribution could act as CCN. This could add a little more nuance to your discussion of climatic effects.

Ziemann, Paul J., Kreidenweis, Sonia M., and Petters, Markus D.. Quantifying the Relationship between Organic Aerosol Composition and Hygroscopicity/CCN Activity. United States: N. p., 2013. Web. doi:10.2172/1086826.

Petters, M.D. and Kreidenweis, S.M., 2007. A single parameter representation of hygroscopic growth and cloud condensation nucleus activity. Atmospheric Chemistry and Physics, 7(8), pp.1961-1971.

Zieger, P., Väisänen, O., Corbin, J.C., Partridge, D.G., Bastelberger, S., Mousavi-Fard, M., Rosati, B., Gysel, M., Krieger, U.K., Leck, C. and Nenes, A., 2017. Revising the hygroscopicity of inorganic sea salt particles. Nature Communications, 8(1), pp.1-10.

AR: The traditional Köhler Theory describes the ability of an aerosol to activate as a CCN and form cloud droplets based on the aerosol's physical and chemical properties. The ability of aerosol to activate as a CCN depends on its chemical composition and size which affect the critical diameter and critical supersaturation. At one time it was thought that the major source of marine CCN originated from secondary sulfate aerosols. It has since been recognized that the ocean also emits primary organic aerosols, which may contribute to marine CCN populations (Quinn and Bates, 2011). However, the contribution of primary marine aerosols to CCN has not been rigorously quantified in part due to the lack of measurements that constrain the amount, chemical composition, and potential sources of organic matter in primary marine aerosols (Brooks and Thornton, 2018). A wide range of marine $\kappa$ values (ranging from a low $\kappa$ value of <0.30 to a high $\kappa$ value >0.90) reported during experiments and field campaigns also suggest that organics may be responsible for this variability.

For multicomponent particles with known components, the $\kappa$ value expressing its CCN activity can be estimated from the $\kappa$ value of each component based on the Zdanovskii, Stokes and Robinson (ZSR) assumption (Petters and Kreidenweis, 2007). Mass fractions were converted to volume fractions using bulk density values. By back-of-the-envelope calculations such as those shown above, the calculated $\kappa$ value of SSA particles in this study is about 0.27 when the largest enrichment factor is considered. An aerosol with a low $\kappa$ value requires a high supersaturation for CCN activation while an aerosol with a high $\kappa$ value needs a lower supersaturation for CCN activation. In relation to the *sc−Dd* curve proposed by (Petters and Kreidenweis, 2007), only 0.1% supersaturation can activate >50% of the particle size distribution as CCN. At 1.0% supersaturation, the entire particle size distribution can act as CCN. However, since the enrichment factor was not calculated by this study model, there may be a large error in the results calculated based on the literature. Therefore, we may not be able to directly estimate the $\kappa$ value of the SSA prepared by us so roughly.

But we have made other changes to the discussion on the climate effect of SSA in the revised manuscript at page 33:

Their team recently developed a process model for understanding the feedback relationship between marine biology, sea spray organic matter, and climate, called OCEANFILMS (Organic Compounds from Ecosystems to Aerosols: Natural Films and Interfaces via Langmuir Molecular Surfactants) sea spray organic aerosol emissions – implementation in a global climate model and impacts on clouds (Burrows et al., 2022).

Furthermore, our results may be an effective complement and development to OCEANFILMS model theory, and by adding the chemical interaction between soluble saccharides and an insoluble fatty acid surfactant monolayer, the consistency of modeled sea spray chemistry with observed marine aerosol chemistry may be improved.

**References**

Brooks, S. D. and Thornton, D. C. O.: Marine aerosols and clouds, Annu. Rev. Mar. Sci., 10, 289-313, 10.1146/annurev-marine-121916-063148, 2018.

Callaghan, A. H., Deane, G. B., Stokes, M. D., and Ward, B.: Observed variation in the decay time of oceanic whitecap foam, J. Geophys. Res.-Oceans, 117, C09015, 10.1029/2012jc008147, 2012.

Christiansen, S., Salter, M. E., Gorokhova, E., Nguyen, Q. T., and Bilde, M.: Sea spray aerosol formation: Laboratory results on the role of air entrainment, water temperature and phytoplankton biomass, Environ. Sci. Technol., 53, 13107-13116, 10.1021/acs.est.9b04078, 2019.

Collins, D. B., Zhao, D. F., Ruppel, M. J., Laskina, O., Grandquist, J. R., Modini, R. L., Stokes, M. D., Russell, L. M., Bertram, T. H., Grassian, V. H., Deane, G. B., and Prather, K. A.: Direct aerosol chemical composition measurements to evaluate the physicochemical differences between controlled sea spray aerosol generation schemes, Atmos. Meas. Tech., 7, 3667-3683, 10.5194/amt-7-3667-2014, 2014.

Cunliffe, M., Engel, A., Frka, S., Gasparovic, B., Guitart, C., Murrell, J. C., Salter, M., Stolle, C., Upstill-Goddard, R., and Wurl, O.: Sea surface microlayers: A unified physicochemical and biological perspective of the air-ocean interface, Prog. Oceanogr., 109, 104-116, 10.1016/j.pocean.2012.08.004, 2013.

Deane, G. B. and Stokes, M. D.: Scale dependence of bubble creation mechanisms in breaking waves, Nature, 418, 839-844, 10.1038/nature00967, 2002.

Fuentes, E., Coe, H., Green, D., de Leeuw, G., and McFiggans, G.: Laboratory-generated primary marine aerosol via bubble-bursting and atomization, Atmos. Meas. Tech., 3, 141-162, 10.5194/amt-3-141-2010, 2010.

Hultin, K. A. H., Nilsson, E. D., Krejci, R., Martensson, E. M., Ehn, M., Hagstrom, A., and de Leeuw, G.: In situ laboratory sea spray production during the Marine Aerosol

Production 2006 cruise on the northeastern Atlantic Ocean, J. Geophys. Res.-Atmos., 115, D06201, 10.1029/2009jd012522, 2010.

Jiang, X. H., Rotily, L., Villermaux, E., and Wang, X. F.: Submicron drops from flapping bursting bubbles, Proc. Natl. Acad. Sci. U. S. A., 119, e2112924119, 10.1073/pnas.2112924119, 2022.

King, S. M., Butcher, A. C., Rosenoern, T., Coz, E., Lieke, K. I., de Leeuw, G., Nilsson, E. D., and Bilde, M.: Investigating primary marine aerosol properties: CCN activity of sea salt and mixed inorganic-organic particles, Environ. Sci. Technol., 46, 10405-10412, 10.1021/es300574u, 2012.

Petters, M. D. and Kreidenweis, S. M.: A single parameter representation of hygroscopic growth and cloud condensation nucleus activity, Atmos. Chem. Phys., 7, 1961-1971, 10.5194/acp-7-1961-2007, 2007.

Prather, K. A., Bertram, T. H., Grassian, V. H., Deane, G. B., Stokes, M. D., DeMott, P. J., Aluwihare, L. I., Palenik, B. P., Azam, F., Seinfeld, J. H., Moffet, R. C., Molina, M. J., Cappa, C. D., Geiger, F. M., Roberts, G. C., Russell, L. M., Ault, A. P., Baltrusaitis, J., Collins, D. B., Corrigan, C. E., Cuadra-Rodriguez, L. A., Ebben, C. J., Forestieri, S. D., Guasco, T. L., Hersey, S. P., Kim, M. J., Lambert, W. F., Modini, R. L., Mui, W., Pedler, B. E., Ruppel, M. J., Ryder, O. S., Schoepp, N. G., Sullivan, R. C., and Zhao, D. F.: Bringing the ocean into the laboratory to probe the chemical complexity of sea spray aerosol, Proc. Natl. Acad. Sci. U. S. A., 110, 7550-7555, 10.1073/pnas.1300262110, 2013.

Quinn, P. K. and Bates, T. S.: The case against climate regulation via oceanic phytoplankton sulphur emissions, Nature, 480, 51-56, 10.1038/nature10580, 2011.

Salter, M. E., Nilsson, E. D., Butcher, A., and Bilde, M.: On the seawater temperature dependence of the sea spray aerosol generated by a continuous plunging jet, J. Geophys. Res.-Atmos., 119, 9052-9072, 10.1002/2013jd021376, 2014.

Sellegri, K., O'Dowd, C. D., Yoon, Y. J., Jennings, S. G., and de Leeuw, G.: Surfactants and submicron sea spray generation, J. Geophys. Res.-Atmos., 111, D22215, 10.1029/2005jd006658, 2006.

Stokes, M. D., Deane, G. B., Prather, K., Bertram, T. H., Ruppel, M. J., Ryder, O. S., Brady, J. M., and Zhao, D.: A Marine Aerosol Reference Tank system as a breaking wave analogue for the production of foam and sea-spray aerosols, Atmos. Meas. Tech., 6, 1085-1094, 10.5194/amt-6-1085-2013, 2013.

Stokes, M. D., Deane, G., Collins, D. B., Cappa, C., Bertram, T., Dommer, A., Schill, S., Forestieri, S., and Survilo, M.: A miniature Marine Aerosol Reference Tank (miniMART) as a compact breaking wave analogue, Atmos. Meas. Tech., 9, 4257-4267, 10.5194/amt-9-4257-2016, 2016.

Textor, C., Schulz, M., Guibert, S., Kinne, S., Balkanski, Y., Bauer, S., Berntsen, T., Berglen, T., Boucher, O., Chin, M., Dentener, F., Diehl, T., Easter, R., Feichter, H., Fillmore, D., Ghan, S., Ginoux, P., Gong, S., Grini, A., Hendricks, J., Horowitz, L., Huang, P., Isaksen, I., Iversen, I., Kloster, S., Koch, D., Kirkevåg, A., Kristjansson, J. E., Krol, M., Lauer, A., Lamarque, J. F., Liu, X., Montanaro, V., Myhre, G., Penner, J., Pitari, G., Reddy, S., Seland, Ø., Stier, P., Takemura, T., and Tie, X.: Analysis and quantification of the diversities of aerosol life cycles within AeroCom, Atmos. Chem.

Phys., 6, 1777-1813, 10.5194/acp-6-1777-2006, 2006.

Wang, X. F., Deane, G. B., Moore, K. A., Ryder, O. S., Stokes, M. D., Beall, C. M., Collins, D. B., Santander, M. V., Burrows, S. M., Sultana, C. M., and Prather, K. A.: The role of jet and film drops in controlling the mixing state of submicron sea spray aerosol particles, Proc. Natl. Acad. Sci. U. S. A., 114, 6978-6983, 10.1073/pnas.1702420114, 2017.

Woolf, D. K., Bowyer, P. A., and Monahan, E. C.: Discriminating between the film drops and jet drops produced by a simulated whitecap, J. Geophys. Res.-Oceans, 92, 5142-5150, 10.1029/JC092iC05p05142, 1987.

Wurl, O., Wurl, E., Miller, L., Johnson, K., and Vagle, S.: Formation and global distribution of sea-surface microlayers, Biogeosciences, 8, 121-135, 10.5194/bg-8-121-2011, 2011.

Xu, W., Ovadnevaite, J., Fossum, K. N., Lin, C. S., Huang, R. J., Ceburnis, D., and O'Dowd, C.: Sea spray as an obscured source for marine cloud nuclei, Nat. Geosci., 15, 282-286, 10.1038/s41561-022-00917-2, 2022.